

# Aerosols, Clouds, and Precipitation in the North-Atlantic Trades Observed During the Barbados Aerosol Cloud Experiment. Part I: Distributions and Variability

Eunsil Jung[1], Bruce A. Albrecht[1], Graham Feingold[2], Haflidi H. Jonsson[3], Patrick Chuang[4], Shaunna L. Donaher[5]

[1]Department of Atmospheric Sciences, RSMAS, University of Miami, Miami, FL, 33149, USA
[2]NOAA Earth System Research Laboratory (ESRL), Boulder, CO, 80305, USA
[3]Naval Postgraduate School, Monterey, CA, 93943, USA
[4]Department of Earth and Planetary Sciences, University of California Santa Cruz, CA, 95064, USA
[5]Department of Environmental Sciences, Emory University, GA, 30322, USA

*Correspondence to*: Eunsil Jung (eunsil.jung@gmail.com)

**Abstract.** Shallow marine cumulus clouds are by far the most frequently observed cloud type over the Earth's oceans; but they are poorly understood and have not been investigated as extensively as stratocumulus clouds. This study describes and discusses the properties and variations of aerosol, cloud, and precipitation associated with shallow marine cumulus clouds observed in the North-Atlantic trades during a field campaign (Barbados Aerosol Cloud Experiment- BACEX, March-April, 2010), which took place off of Barbados where African dust periodically affects the region. The principal observing platform was the Center for Interdisciplinary Remotely Piloted Aircraft Studies (CIRPAS) Twin Otter (TO) research aircraft, which was equipped with standard meteorological instruments, a zenith pointing cloud radar and probes that measured aerosol, cloud, and precipitation characteristics.

The temporal variation and vertical distribution of aerosols observed from the 15 flights, which included the most intense African dust event during all of 2010 at Barbados, showed a wide range of aerosol conditions. During dusty periods, aerosol concentrations increased substantially in the size range between 0.5 µm and 10 µm (diameter), particles that large enough to be effective giant cloud condensation nuclei (CCN). The 10-day back trajectories showed three distinct air masses with distinct vertical structures associated with air masses originating in the Atlantic (typical maritime air mass with relatively low aerosol concentrations in the marine boundary layer), Africa (Saharan Air Layer), and mid-latitudes (continental pollution plumes). Despite the large differences in the total mass loading and the origin of the aerosols, the overall shapes of the aerosol particle size distributions were consistent, with the exception of the transition period.

The TO was able to sample many clouds at various phases of growth. Maximum cloud depth observed was less than ~ 3 km, while most clouds were less than 1 km deep. Clouds tend to precipitate when the cloud is thicker than 500-600 m. Distributions of cloud field characteristics (depth, radar reflectivity, Doppler velocity, precipitation) were well identified in the reflectivity-velocity diagram from the cloud radar observations. Two types of precipitation features were observed for



shallow marine cumulus clouds that may impact boundary layer differently: first, a classic cloud-base precipitation where precipitation shafts were observed to emanate from the cloud base; second, cloud-top precipitation where precipitation shafts emanated mainly near the cloud tops, sometimes accompanied by precipitation near the cloud base. The second type of precipitation was more frequently observed during the experiment. Only 42-44 % of the clouds sampled were non-

precipitating throughout the entire cloud layer and the rest of clouds showed precipitation somewhere in the cloud, predominantly closer to the cloud top.

**Keywords**: African dust outbreak, cloud-top precipitation, Saharan Air Layer (SAL), North-Atlantic trades, Barbados

**1 Introduction**

Shallow marine cumulus clouds are frequently observed over the Earth's oceans and are by far the most common type of cloud in the world (Norris, 1998). The fractional cloudiness associated with these cumulus clouds is typically 15 to 25 %, but the extensive areas that the shallow cumuli cover make their radiative impact an important factor in the climate system. Further, shallow cumulus clouds are part of the feeder system for deep convection in the tropics and are critical to the energy

and moisture budget of the trade wind boundary layer. Recent studies indicate that these clouds give the largest uncertainty in tropical cloud feedbacks in the climate system (e.g. Bony and Dufresne, 2005; IPCC, 2013) and therefore must be better understood.

The earliest field programs—the 1969 Atlantic Trade-Wind Experiment (ATEX) and the 1969 Barbados Oceanographic and Meteorological Experiment (BOMEX)—provided rawinsonde data sets that were used to estimate

enthalpy and moisture budgets associated with shallow, undisturbed cumulus clouds (Augstein et al., 1973; Holland and Rasmusson, 1973). Although several aircraft were associated with BOMEX (Friedman et al., 1970), the instrumentation on these aircraft was not adequate for routine measurements of cloud properties. In the 2004-2005 Rain in Cumulus over the Ocean (RICO; Rauber et al., 2007), instrumented aircrafts were used to sample clouds and precipitation and key processes operating in these clouds observed over the Eastern Caribbean (e.g. Hudson et al., 2007; Colon-Robles et al., 2006; Gerber et

al., 2008). Further, the Barbados Cloud Observatory (operated by the Max Planck Institute for Meteorology, Stevens et al., 2015), using ceilometer, Raman lidar, and cloud radar observations, has provided a 2-year climatology of non-precipitating and precipitating cumulus (Nuijens et al., 2014), and the relative influences of aerosols and meteorology on precipitation formation (Lonitz et al., 2015). In the same area, detailed aerosol, cloud, radiation, and turbulence observations were made from a platform suspended from a helicopter operating in an area off the coast of Barbados as part of the CARRIBA (Cloud,

Aerosol, Radiation and tuRbulence in the trade wind regime over BArbados) project (Siebert et al., 2013).

The marine environment near Barbados provides an excellent area to sample shallow marine clouds that have a strong propensity to precipitate. In addition, African dust outbreaks periodically affect the clouds over the regime, and offer



an excellent opportunity to observe aerosol-cloud-precipitation interactions. Furthermore, near surface aerosol measurements have been made on the island since the 1960's (Prospero and Lamb, 2003). To better understand aerosol-cloud-precipitation interactions in the trade cumuli regime, Barbados Aerosol Cloud Experiment (BACEX) was carried out off the Caribbean island of Barbados, within the northeast trades of the North-Atlantic from mid-March and mid-April 2010 (Jung et al., 2013).

The goal of the BACEX study is to improve our understanding of aerosol-cloud-precipitation processes in the trade-wind cumulus regime, and thus, to improve and/or provide a basis for evaluating and improving the parameterization of cloud-aerosol-precipitation interactions in numerical models. As a first step, this paper is intended to document the properties of shallow marine cumulus clouds and the vertical structure of the Saharan Air Layer (SAL), and provide reference data for interpreting and comparing satellite data. The findings from this study confirm some previous results and also add new
insights to the distribution and variability of clouds and aerosols in the North-Atlantic trades. The interactions among aerosol, cloud and precipitation will be addressed in a separate study, and thus, cloud and precipitation responses to the aerosols, including cloud particle size distributions, are not discussed in the current paper.

Satellite-based studies have been used to examine aerosol-cloud interactions over large geographical areas for extended time periods, but are known to suffer from retrieval biases (Loeb and Schuster, 2008) and the vertical distribution –
a key component of the aerosol – is usually unknown. Thus, we combine the in situ aircraft data from BACEX with soundings from the island to explore the boundary layer structure and properties of clouds and aerosols over this area of the Caribbean. Data used in this study and methods are described in Sect. 2. The overall large-scale atmospheric conditions during the experiment, aerosol source-regions observed at Barbados, temporal and vertical variations of aerosols are discussed in Sect. 3, along with cloud and precipitation properties including radar reflectivity and Doppler velocity
distribution of clouds, two types of precipitation (classic cloud-base precipitation versus cloud-top precipitation), non-adiabatic characteristics of shallow marine cumulus clouds, and followed by the summary and discussion in Sect. 4.

## 2 Data and methods

The Center for Interdisciplinary Remotely-Piloted Aircraft Studies (CIRPAS) Twin Otter (TO) research aircraft made 15 flights from 19 March to 11 April upstream of Ragged Point (13.2 °N, 59.5 °W) on the eastern shore of Barbados (see Fig. 1
in Jung et al., 2013 for the location), which has a long history of surface aerosol measurements (Prospero and Lamb, 2003). Each flight was of 3-4 hour duration and included at least one pseudo sounding made as the aircraft either ascended or descended, and several horizontal level flights from near the ocean surface to above the trade-wind inversion. The common horizontal level flight patterns included measurements (1) near the ocean surface (30 m level leg), (2) in the sub-cloud layer (between cloud layer and ocean surface), (3) near cloud base height, and (4) at cloud-top (the maximum height at which the
aircraft still encountered cloud). General information on individual flights is shown in Table A1 in the appendix. The feasibility of using a passive tracer in the form of radar chaff was explored on some of the flights to study entrainment and transport processes in small cumuli (Jung and Albrecht, 2014) and is also noted in Table A1.



### 2.1 Aircraft data

The TO research aircraft was equipped with probes that measure aerosol, cloud and precipitation in addition to the standard meteorological instruments for observing the mean and turbulent thermodynamic and wind structures as described in Zheng et al. (2011) and Jung et al. (2013). The standard meteorological variables (e.g., temperature, water vapor mixing ratio, winds) and PVM-100 water content (Gerber et al., 1994) were obtained at 10 Hz and then averaged to 1Hz. Aerosol data included aerosol number concentration ($N_a$) from a Passive Cavity Aerosol Spectrometer Probe (PCASP, 0.1-2.5 µm), cloud condensation nuclei (CCN) from a CCN spectrometer (Droplet Measurement Technologies inc.), and condensation nuclei from the Condensation Particle Counters (CPCs). Cloud and precipitation data included cloud drop number concentration ($N_d$) from the Cloud Aerosol Spectrometer (CAS, 0.6-60 µm) and drizzle water contents from the Cloud Imaging Probe (CIP, 25-1550 µm). Aerosol concentration (CN, CCN) and probe data (e.g., PCASP, CAS, and CIP) were also obtained at 10 Hz and then averaged to 1 Hz.

Vertically pointing cloud radar (95 GHz, bistatic, Frequency Modulated Continuous Wave Doppler radar) was mounted on top of the aircraft fuselage and detected fine vertical structures of updrafts and downdrafts within the clouds and precipitation properties. The radar data were obtained at a sampling rate of 3 Hz with range gates of 24 m, an antenna beam width of 0.7 °, a velocity resolution of 0.16 m s$^{-1}$, and a dead zone of less than 50 m (Jung and Albrecht, 2014), allowing radar observations to be made in close proximity to the in situ probe measurements. The instruments used for this study are summarized in Table 1. Acronyms and symbols are listed in Table A2 in the appendix.

### 2.2 Ragged Point aerosol measurements

Near-surface aerosol measurements were made at Ragged Point, a site located on the edge of a 30 m high bluff on the easternmost coast of Barbado. Samples were collected from the top of a 17 m high tower using a high-volume filter sampling system. Filters were changed daily and periodically returned to Miami where the soluble components were extracted with water (Li-Jones et al., 1998). The Ragged Point aerosol site is operated by Dr. Joseph Prospero's research group from the University of Miami.

### 2.3 Large-scale conditions

The large-scale time-height variability of temperature, moisture, and wind structures in the study region was defined using observations from rawindsondes launched at Grantley Adams airport (13.06 °N, 59.48 °W, WMO ID: 78954) on Barbados. Sounding data were obtained from the University of Wyoming's online upper-air data (http://weather.uwyo.edu/upperair/sounding.html). Further atmospheric conditions were obtained from soundings made by the TO during ascents and descents of the aircraft.



## 2.4 Back-trajectories

The origin of air masses sampled by the TO was estimated using the Hybrid Single Particle Lagrangian Integrated Trajectory (HYSPLIT; http://ready.arl.noaa.gov/HYSPLIT_traj.php) model using an average location of the flight domain (13.2 °N, 59 °W, Fig. 1 in Jung et al., 2013). The 10 day back trajectories, arriving at Barbados at 500 m above ground level (AGL), were calculated to give a general sense of the air mass source regions sampled on aircraft missions in the sub-cloud layer.

## 2.5 Aerosol particle size and distribution

An effective aerosol particle size ($D_a$) and its size distribution (PSD) were estimated from the accumulation mode aerosol that was measured by PCASP in cloud-free air (i.e., flight-averaged aerosol particle size and distribution; e.g., Fig. 6). In addition to the flight-averaged characteristics, $D_a$ and PSDs in the sub-cloud layer were calculated by combining PSDs from PCASP (up to channel 19) and CAS (from channel 10 and up) probes, to give the full size and distribution information, for the range from 0.1 µm to 60 µm. The PCASP (0.1 – 2.5 µm) dries the particles before measuring them, while CAS (0.6 – 60 µm) sizes them in ambient conditions. This could result in discrepancies between the two values especially when the ambient relative humidity (RH) is high (e.g., peaks near 2-3 µm in Fig. 7, later). Nevertheless, the two size-distributions line up well at the interface when the ambient RH is less than 70-80 %. $D_a$ was calculated as in Eq. (1).

$$D_a = \frac{\int N(D)D^3 dD}{N(D)D^2 dD} ,$$ (1)

where D was the bin-mean diameter of the probe.

## 2.6 Rainfall rate

Rainfall rate (mm h$^{-1}$) was calculated from the CIP drop size distribution (e.g., Rogers and Yau, 1989) using

$$R = \frac{\pi}{6} \int_{25\mu m}^{1550\mu m} N(D)D^3 u(D)dD ,$$ (2)

where $u(D)$ was the fall speed of a drop with diameter $D$. Here, three fall speed formulations were used for differing drop sizes: (1) $u = k_1 r^2$ was used for cloud droplets up to 30 µm radius with $k_1 \approx 1.19\times10^6$ cm$^{-1}$ s$^{-1}$; (2) $u = k_2 r^{1/2}$ was used with $k_2 \approx 2.01\times10^3$ cm$^{1/2}$ s$^{-1}$ for droplets in the radius range of 0.6 mm $< r <$ 2 mm; and (3) $u = k_3 r$ with $k_3 \approx 8\times10^3$ s$^{-1}$ was used for the intermediate size range of 40 µm $< r <$ 0.6 mm.





## 3 Results

### 3.1 Large-scale atmospheric conditions

General features of large-scale atmospheric conditions over the study area are shown in Fig. 1 by time-height cross-sections of humidity, temperature, wind speed, and wind direction from the Barbados soundings for the period of 14 March and 16 April 2010. During the experiment, the Lifting Condensation Level (LCL; calculated from the average thermodynamic properties of the sub-cloud layer) was lower than 1 km (~747 m on average, which agreed with the two-year climatology of LCL in this region as documented by Nuijens et al., 2014) and the 0 °C isotherm was near 5 km. Since the maximum cloud depth was less than 3 km AGL (Fig. 13 shown later), the clouds during the experiment were warm (liquid phase only). The inversion height (square and cross symbols) increased from ~ 1.5 km to 3.7 km from 18 March to 25 March, and then decreased to a minimum of ~ 1 km on 3-4 April as dry air intruded into the lower atmosphere. After 5 April, the inversion height increased and the lower-troposphere stability weakened (not shown). The primary inversion height here was defined as the level below 6 km where the increase in virtual potential temperature with height was the greatest over a 5 m interval. A secondary maximum was also identified. The appearance of multiple inversion heights was common during the experiment. The variations in inversion heights agreed with the changes in vertical structures of winds, humidity and temperature in Fig. 1.

Atmospheric humidity conditions (Fig. 1a) showed significant dry air intrusions into the layer below 2 km, prior to 22 March and from 31 March to 5 April (dusty period). On 5 April, a sharp dry-to-moist transition occurred through the entire lower atmosphere. The sub-cloud layer (below the LCL) was relatively well-mixed throughout the field experiment (Fig. 1b), showing a constant θ at ~ 300 K. Prior to 31 March (pre-dust outbreak periods), easterlies dominated throughout the atmosphere. The heights of the easterlies lowered with the onset of dust outbreaks on 31 March, and then reached a minimum height of ~ 1 km on 3-4 April when the inversion heights were the lowest and the air mass was the driest, which corresponds to the period of African dust event observed at Barbados. After 4 April, the regions of easterlies ascended and the maximum easterlies appeared at around 4-5 km on 9-10 April when the lower atmosphere experienced the coldest (not shown) and the most humid conditions. Similarly, anomalously weak winds were noticeable during the dust outbreak (Fig. 1c). The regions of weak winds (e.g. < 5 m s$^{-1}$) descended from 3-5 km on 31 March to 1-2 km during the dust outbreak (31 March-5 April).

There was no precipitation recorded at the surface weather station in Barbados during the campaign (no rain or trace recorded, http://www.wunderground.com/global/BR.html). The mean precipitable water during BACEX was 4.1 cm based on the soundings at Grantly Adams airport (not shown), and was similar to observations during the RICO field campaign (Rauber et al., 2007).

The overall atmospheric conditions and variability observed from the TO are illustrated in Fig. 2 with the vertical profiles of potential temperature ($\theta$), mixing ratio ($q$), and aerosol concentration ($N_a$). Only one sounding from each flight is shown. Potential temperature (Fig. 2a) varies from 298 K to 312 K with poorly defined inversion heights in most flights.





However, strong inversion layers were observed during RF10 and RF11 (1-2 April, yellow and light orange) near 500-600 m and 1500-1700 m that were characterized by θ jumps and significant reductions in $q$. RF1 (19 March) also showed a strong inversion near 1700 m, along with a significant reduction in $q$ near the layer. The overall $N_a$ for these profiles (RF1, RF10, and RF11) was relatively high, compared with other days. In contrast, profiles from RF13, RF14, and RF15 showed a monotonic increase in θ and a decrease in $q$ with height, without any significant inversions or dry layers. The $N_a$ was relatively low on these days (Fig. 2c) with concentration of less than 250 mg$^{-1}$ below 1000 m that decreased monotonically with heights to ~0 mg$^{-1}$ above 1500 m.

To determine how the BACEX thermodynamic structures compare with those obtained from previous field campaigns in Caribbean cumulus regimes, we compare these structures with the vertical profiles of θ and $q$ obtained from BACEX, ATEX, BOMEX and RICO. All the data sampled, including soundings and level flights, were used to attain the mean profiles of θ and $q$ for BACEX. Further, the data were first averaged at 20 m vertical intervals, and then subjected to a 9-point moving average (87.5 m resolutions) to filter out small variations. θ and $q$ for RICO and BOMEX were obtained from GCSS (GEWEX Cloud System Study) boundary layer cloud homepage (http://www.knmi.nl/~siebesma/BLCWG/), and Table 1 from Stevens et al. (2001) for ATEX. Some of the GCSS thermodynamic soundings are simplified realizations of the detailed average soundings.

In Fig. 3, BOMEX shows similar moisture conditions as BACEX below the inversion (~ 1500 m), but is drier than BACEX above the inversion by about 5 g kg$^{-1}$. θ during BOMEX is about 1 K cooler (warmer) than BACEX below (above) the inversion, but is within ±1σ. The RICO profile shows consistently cooler (~2 K) and drier (~2 g kg$^{-1}$) atmospheric conditions than BACEX throughout the boundary layer (except between 1000 – 1300 m). The cooler and drier conditions during RICO compared with BACEX are due to the time of year and latitude differences between the two field projects. The ATEX profile shows the driest and coldest conditions amongst the others. θ during ATEX is about 4 K cooler than BACEX below the inversion and about 1 K warmer above the inversion (~ 1500 m). Further, $q$ during ATEX is about 4 g kg$^{-1}$ lower than $q$ during BACEX. These differences can be attributed to the higher latitude (lower SST conditions) of the ATEX observing area. Overall, atmospheric conditions during BACEX (March-April, 2010) were warmer and moister than the others but similar to those from CARRIBA made during dry months (CARRIBA$_{DRY}$; Fig. 5 in Siebert et al., 2013) that took place in a similar area and time of the year (i.e., spring).

**3.2 Back trajectories**

Back trajectories were calculated to give a rough indication of the air mass source regions observed in the boundary layer during BACEX. The 10 day backward trajectories were calculated by using daily 12 UTC air masses, observed at 500 m in the middle of the flight domain (Fig. 4). The air mass within the boundary layer over Barbados (Fig. 4) originated mainly from three regions, in agreement with the findings of Dunion (2011). The first group of similar air-mass source-regions occurred on 19 March and during 30 March and 5 April. They corresponded to the periods of dry air intrusion into the lower troposphere (Fig. 1a) when the air mass originated from Africa (dust outbreak period). The second group of similar air mass



source-regions occurred between 23 and 26 March, and originated from middle latitudes continents (e.g., North America). The third group (e.g., 3/22, 3/29, 4/10, 4/11) originated from the North Atlantic with trajectories remaining over the ocean for at least 10 days.

## 3.3 Aerosol properties

### 3.3.1 Vertical and temporal variation

African dust events across the North Atlantic, including the period of BACEX, suggested a series of SAL outbreaks (not shown). Prior to BACEX, a large SAL event occurred on 16 March from the African coast. Over the next few days, dust spreads over the North Atlantic. Then, another surge of dust occurred over Africa on 22, and 25-26 March based on the satellite images and vertical structures of $\theta$ and $q$ over Africa. The dust event observed at Barbados between late March and early April (Fig. 4, Fig. 5c) was mainly a result of these surges of dust (Jung et al., 2013). During BACEX, a wide range of aerosol conditions was observed on 15 flights, including the most intense African dust event observed at the Barbados surface site during all of 2010.

Aerosol concentrations measured at the surface and in the sub-cloud layer are shown in Fig. 5, along with the vertical structures of aerosol concentration in the trade-wind boundary layer. Dust surface concentration (Fig. 5c) was obtained at the Ragged Point surface site. Sub-cloud $N_a$ (Fig. 5b) was obtained from TO during the level leg flights in the sub-cloud layer; and vertical profiles of $N_a$ (Fig. 5a) were obtained from TO during the aircraft's ascents and/or descents. In Fig. 5c, dust surface mass concentration remained lower than 10 µg m$^{-3}$ prior to 29 March, and then rapidly increased to a maximum on 1-2 April with mass concentration exceeding 150 µg m$^{-3}$. The aerosol robotic network (AERONET; Holben et al., 1998) level 2 Aerosol Optical Depth (AOD) at 500 nm wavelength fluctuated around 0.1 during the non-dusty period, and then increased rapidly from 29 March to 1 April, when AOD was observed to be ~ 0.6.

Temporal variations of sub-cloud aerosols are shown in Fig. 5b. The mean values of CN (black), PCASP (blue) and CCN (activated at a super-saturation of 0.6 %, hereafter $s$=0.6 %, red) are shown as solid lines with ±1σ denoted by vertical error bars. Overall, CCN concentration followed PCASP (i.e., accumulation aerosol mode) patterns reasonably well. The aerosol concentration showed an increasing trend from 29 March to 5 April for CN, PCASP and CCN, which was consistent with the trend of dust surface concentration in Fig. 5c. In contrast, high aerosol concentration in the sub-cloud layer on 23 March (Fig. 5b) was not from African dust; but may have originated over the East coast of North America based on the back trajectories (Fig. 4).

The vertical distribution of $N_a$ is shown in Fig. 5a. The average (black-solid lines) and individual (colored lines) profiles of $N_a$ (m g$^{-1}$) are offset by 400 mg$^{-1}$ for each flight in Fig. 5a, with each vertical dotted line representing a new axis to indicate aerosol concentration for the day in question. For example, $N_a$ on 5 April is nearly constant below 500-600 m (~ 300 mg$^{-1}$), then gradually increases with height and peaks around 2000 m at ~ 600-700 mg$^{-1}$. Thereafter $N_a$ decreases with



height reaching ∼ 200 mg$^{-1}$ at around 2500 m. Measurements of $N_a$ on 23 March were not available, thus CCN ($s$=0.6 %) are overlaid in Fig. 5a to give a general sense of $N_a$ vertical structure on the day.

The variety of vertical structures, evident in Fig. 5a, is of interest; $N_a$ decreases monotonically as height increases on 22, 29, 30 March, and 7, 10, 11 April with a maximum $N_a$ in the sub-cloud layer. In particular, observations on 22, 29 March and 10, 11 April show relatively clean boundary layer conditions with a maximum $N_a$ of less than 200 mg$^{-1}$. These air masses appear to originate over the Atlantic (Fig. 4). In Fig. 5, there are a couple of days when high $N_a$ is observed above the inversion, but low $N_a$ is recorded at the surface (e.g., 25, 26 March). Both days show high AOD in Fig. 5c, suggesting that AOD may not be a good indicator of the low boundary layer aerosols, which are important to low-level clouds. Air masses on these days (25-26 March) originated from mid-latitudes continents (Fig. 4). During the period between 31 March and 5 April (dusty periods), there was significant aerosol variability in the marine boundary layer (Fig. 5a). The vertical structures of aerosols and their source regions are summarized in Table A3 in the appendix.

### 3.3.2 Aerosol particle size distribution

The BACEX aircraft observations provided a characterization of the variability in the aerosol particle size distributions (PSDs) (Fig. 6). PSDs were calculated from all available PCASP measurements made on pseudo-soundings and level flights for a given day, when no liquid water was detected, to give daily flight-averaged PSDs. The daily flight-averaged PSDs have a maximum concentration at about 0.15 μm in the fine mode (Fig. 6). It is of interest that the overall shapes of PSD are consistent, despite the large differences in the total mass loading and the origin of aerosols (see Figs. 4 and 5). Nevertheless, slight differences are observed between the individual PSDs, and those differences provide some insights into the processes (such as cloud processing of the aerosols, see Jung et al., 2013) that affect the aerosol concentration. For example, PSDs obtained from RF07 (3/29; green bold-solid) and RF08 (3/30; green bold-dashed) have a similar concentration for the smaller sizes of aerosols (e.g., D < 0.25 μm), but the difference increases as the aerosols increase in size with more abundant larger particles as observed on 30 March. Although these two days have similar vertical structures in $N_a$ (Fig. 5a), the air masses sampled on 30 March originated from Africa (dust particles), whereas air masses sampled on 29 March originated from the Atlantic (sea salt) (Fig. 4). Sea salt particles on 29 March could serve as giant CCN (GCCN), but GCCN concentrations in nature are many orders magnitude less than CCN concentrations (order 10$^2$ cm$^{-3}$, Fig. 5b) (Feingold et al., 1999), and thus, not likely to contribute to the larger sizes of aerosols in Fig. 6. On 31 March (RF09; light green bold-solid), African dust prevailed throughout the trade-wind boundary layer, and $N_a$ increased over all sizes. PSDs on 25-26 March (RF05-06), which originated from mid-latitudes continent (Fig. 4), showed large aerosol loadings above the inversion (Fig. 5) reflecting the abundance of smaller particles and lack of larger particles on these days, implying different aerosol sources from the dusty periods.

To include larger particle sizes, PSDs are calculated by combining PSDs obtained from PCASP with CAS in the sub-cloud layer (Fig. 7). The plots in Fig. 7 show two distinct populations of PSDs in the sub-cloud layer. First, PSDs from dusty days (RF07-RF12) have a significantly higher $N_a$ and total volume between 0.5 μm and 10 μm, compared with PSDs





that were obtained from the non-dusty days. The increase in $N_a$ in this particular size-range may have important impacts on cloud-aerosol interactions because the most effective GCCN in terms of particle size lie within this range (diameters of 3-6 μm are optimal for enhancing precipitation in warm clouds (See Segal et al., 2004; Jung et al., 2015). However, since pure dust is insoluble, to serve as a GCCN the dust particle would need to be coated with hygroscopic materials. An example of

dust particles playing a role as a GCCN is shown in Levin et al. (2005) during the Mediterranean Israeli Dust Experiment Campaign. Remaining PSDs – the second distinct populations of PSDs in the sub-cloud layer – are associated with non-African dust periods and show relatively low $N_a$ over all size ranges. The effect of measuring the size of dust versus salt, which have different refractive indices, is relatively small in PCASP. In the case of CAS, however, uncertainties in sizing particles smaller than about 10 μm, can be as much as a factor of two (not shown). In the combination of PCASP and CAS

(e.g., Fig. 7), the first four combined bins of CAS (channels 10 to 13) are likely subject to size uncertainties due to refractive index differences between dust and salt.

**3.4 Cloud and precipitation properties**

During the experiment, small cumulus clouds were observed on most days, whereas relatively deep cloud clusters (heights to about 2.5-3 km) were sampled on only a few days (e.g., 22, 24, and 30 March) with different characteristics relative to the

small cumulus clouds. The cloud radar data sampled during the cloud-base level-leg flights were used to obtain a bulk sense of shallow marine cumulus cloud characteristics, such as distributions of cloud reflectivity, velocity, thickness, tops and bases. The dates, time periods, and average heights of level-leg flights used for the radar analysis are summarized in Table A4. Examples of time-height cross-sections of radar reflectivity, for the 5-minute periods, are shown in Fig. 8. Radar reflectivity z is written as (3) by assuming particles are spherical and small compared with the radar wavelength

$$z = \int N(D)D^6 dD \qquad (3)$$

in units of $mm^6\,m^{-3}$. Throughout the text, radar reflectivity $Z$ in units of dBz, is used as radar reflectivity, where $Z=10\log(z)$.

The clouds sampled on 22 and 24 March (Figs. 8a-b) were precipitating, and characterized by strong reflectivity (e.g., $Z > -20$ dBz) and larger physical sizes (horizontal and vertical). On the other hand, on 29 March and 11 April (Figs 8c-d), typical marine shallow cumulus clouds were sampled that produced substantially weaker reflectivity ranging from -40

dBz to -20 dBz. These non-precipitating-clouds are narrower (less than 1 km wide) and shallower (depths less than 500 m) than the precipitating-cloud systems that are about 4-7 km wide and 1-2 km depth (Fig. 8a-b). Further, precipitating clouds tend to exhibit more organized mesoscale features, and the hydrometeor reflectivity is high enough to be detected by the S-band radar located on Barbados (not shown).

To show the organizational differences between precipitating and non-precipitating clouds, satellite imagery taken

on 22, 24, 29 and 30 March is shown in Fig. 9. Clouds on 22 March comprise relatively deep convective cores surrounded by cloudiness that is formed from the outflow of the deeper convection (Fig. 9a). The clouds appear to be organized around the arc-shaped outflow boundaries from earlier convection as shown in the RICO field campaign (Zuidema et al., 2012), and this





organizational characteristic is also evident on 24 and 30 March (Fig. 9b and Fig. 9d). Convection associated with these features often reached cloud heights of about 2-3 km. On the other hand, the aircraft sampled typical fair weather cumulus clouds on 29 March (Fig. 9c). The size of the clouds was significantly smaller than the precipitating cloud systems, and clouds did not have outflow features as did in the precipitating clouds. This shallow convection had no measurable

precipitation, and often had a cloud thickness of less than 500 m (Fig. 13 shown later).

Characteristics of cloud cores sampled during BACEX are shown in Fig. 10. A cloud core was defined by updrafts ($w$) greater than 1 m s$^{-1}$. The adiabatic liquid water mixing is overlaid on Fig. 10a. The 10 m vertically averaged liquid water mixing ratio (Fig. 10a) and cloud droplets number concentration ($N_d$) (Fig. 10b) for non-precipitating clouds were estimated using data with $w > 1$ m s$^{-1}$, PVM-100 LWC $> 0.01$ g m$^{-3}$, and CIP volume number concentration $< 0.01$ cm$^{-3}$. The criterion

for CIP volume number concentration is applied here because shattering of large drops can contaminate the measurements of $N_d$ and also large drops tend to precipitate. Overall, clouds sampled below 1 km are mainly non-precipitating and close to adiabatic, while clouds observed above 2.2 km are mostly precipitating. Figure 10(a) shows that shallow cumulus clouds sampled during BACEX are far from adiabatic even in the cloud core, which is in agreement with marine cumulus clouds sampled during RICO (e.g., Rauber et al., 2007; Gerber et al., 2008) and continental cumulus clouds (Lu et al., 2008).

BACEX was seasonally similar to the CARRIBA$_{DRY}$ period but observed the strongest dust event of 2010 as well as more typical marine boundary layer aerosol, and provided at least three different types of aerosols: i) Sea salt during typical maritime air masses, ii) dust particles during the dust events, and iii) fine particles from long-distance continental pollution plumes mainly residing above the boundary layer, giving the wide range of $N_d$ that was not seen in other studies experiencing relatively homogeneous aerosols environments (e.g., Gerber et al., 2008). $N_d$ varies from near 0 to 400 cm$^{-3}$ and tends to

increase with height (Fig. 10b). The low $N_d$ at high altitude (~ 2300 m) may be associated with entrainment mixing and wet scavenging due to precipitation.

To characterize the cloud and precipitation structures observed during BACEX, radar reflectivity $Z$ and Doppler velocity $V_r$ are examined in Fig. 11 as the frequency distributions of $Z$ versus $V_r$. Here $Z$ and $V_r$ are measured from the vertically pointing cloud radar during the cloud-base level-leg flights (except for 30 March). The $V_r$ is the sum of the

25 hydrometeor fall velocity, $V_f$ (e.g., raindrop) and the air motion, $w$ ($V_r = V_f + w$), and positive $V_r$ indicates updrafts. In Fig. 11, clouds sampled on 25, 26, 29, 31 March, and 10, 11 April all show similar patterns in $Z$ and $V_r$. On the $V_r$ -$Z$ diagram, frequency distributions are horizontally oriented, indicating a wide range of $V_r$ (-5 to 3 m s$^{-1}$) and a narrow range of weak $Z$ (-38 dBz ~ -28 dBz). In contrast, the second group (22, 24, 30 March, and 7 April) shows $V_r$ and $Z$ that are vertically oriented on the diagram with a relatively narrow range of negative $V_r$ and a wider range of weak to strong $Z$. In particular, clouds

sampled on 22, 24, and 30 March show the maximum frequency of $Z$ stronger than -20 dBz. Clouds sampled on 23 March and 5 April show a mix of both types of distributions. Clouds with weak $Z$ tend to be non-precipitating. In contrast, clouds with a maximum frequency appearing at strong Z (e.g., $Z > -20$ dBz here, also in Frisch et al., 1995) and negative $V_r$ are generally precipitating (e.g., 22, 24 and 30 March in Fig. 11). Clouds sampled on these three days (3/22, 3/24, and 3/30) are



referred to as "precipitating-clouds" hereafter, whereas clouds sampled on 9 days that excluded these three precipitating-cloud cases are referred to as "non-precipitating clouds".

Cloud composites are shown in Fig. 12 on the $V_r$ –$Z$ diagram. During BACEX, clouds with $Z$ between -20 dBz and 5 dBz and $V_r$ between -2 m s$^{-1}$ and 1 m s$^{-1}$ are most frequently sampled (Fig. 12a; number of samples > 200). However, this

peak is strongly influenced by precipitating-clouds, and is evident in the precipitating-cloud composite in Fig. 12b that exhibits a similar distribution of the cloud composite made from the entire BACEX periods (Fig. 12a). The $V_r$ -$Z$ distribution, estimated by excluding the strongest precipitating cloud on 22 March (Fig. 12c), shows two populations of $Z$ and $V_r$: first, $Z$ less than -30 dBz and $V_r$ ranging from -4 m s$^{-1}$ to 3 m s$^{-1}$ (horizontally oriented pattern in Fig. 12c); and second, $Z$ ranging from -30 dBz to 5 dBz and $V_r$ ranging from 0 m s$^{-1}$ ~ -4 m s$^{-1}$ (vertically oriented pattern in Fig. 12c). $V_r$–$Z$

distribution that excludes the three precipitating-clouds shows a horizontally oriented pattern in Fig. 12d with a wide range of $V_r$ and narrow range of weak $Z$ (-40 ~ -30dBz), confirming that those clouds are predominantly non-precipitating. However, Fig. 12d also shows the other regime of $Z$ and $V_r$ (vertically oriented pattern) that indicates the presence of lightly precipitating clouds in the shallow marine cumulus cloud regimes that are dominated by non-precipitating clouds. The cloud thickness of these non-precipitating clouds (but with light precipitation, Fig. 12d) is about 1300 m (Fig. 13c-d).

Vertical frequency distributions of $Z$ and $V_r$ are shown in Fig. 13. Two dominant populations of $Z$ are shown in Fig. 13a, which are composed of all of the data – one with $Z$ < ~ -35 dBz at around 1000 m, and the second with $Z$ > -20 dBz between 1000 and 2300 m. For the same composite of clouds, the velocity distribution (Fig. 13b) peaks at about -2 m s$^{-1}$ to 0.5 m s$^{-1}$ between 1000 and 2300 m. Cloud bases and tops are about 400 m and 2700 m, respectively, indicating a maximum depth of the clouds of about 2300 m. A striking feature in Fig. 13a-b is the jump from small $Z$ to large $Z$ populations over a

short vertical distance near 1000 m (Fig. 13a), showing clouds deeper than 500-600 m (in depth) have a significant chance of raining. A similar behavior of trade-wind cumuli has been noted in the early work of Malkus (1958). $Z$ and $V_r$ frequency distributions of the 9 days, excluding three precipitating clouds, are shown in Fig. 13(c-d). For these non-precipitating clouds, $Z$ ~ -35 dBz and $V_r$ of ± 2 m s$^{-1}$, are the most frequently observed between 600 m and 1300 m. Cloud bases and tops for these clouds are about 700 m and 2000 m, respectively, indicating a thickness of about 1300 m.

The vertical structures of the individual clouds are further examined in Fig. 14. For a given day, the total number of data points at a given height is counted based on data sampled along the cloud-base level leg flights by the cloud radar. Then, the number of data points is divided by the maximum number of each day to have the same range from 0 and 1. This approach is to facilitate comparisons with other days, since the main purpose of this calculation is to examine the differences in vertical sampling statistics between individual days, in particular between precipitating and non-precipitating clouds. Here

in Fig. 14, the terminology "clouds" is used for the area and/or data points where the cloud radar detects signals. We assume that an individual observation represents a precipitating cloud if $Z$ > -20 dBz and $V_r$ < 0 based on Figs. 11-13. The data shown in Fig. 14 are averaged in 100 m vertical intervals to filter out small variations.

Two types of precipitating-clouds are shown in Fig. 14. The first cloud type has precipitation shafts that are observed mainly close to and below the cloud base (and/or throughout the most of the cloud layer; this feature is also seen in





Fig. 13b with stronger downward motions close to cloud bases), especially when the clouds are deeper than the other lightly precipitating clouds. For example, on 22 March, the overall occurrence of precipitating clouds (black) exceeds the occurrence of non-precipitating clouds (grey) close to the cloud base. In addition, the height of maximum occurrence of the precipitating clouds is slightly lower than the height of maximum occurrence of total clouds. The second precipitating cloud
type has precipitation shafts that emanate mainly from the upper part of the cloud and/or near cloud top (e.g., 3/24, 4/10 in Fig. 14; hereafter cloud-top precipitation) on the downshear side of the cloud (not shown). This type of precipitating cloud is shallower than the first type of cloud, and can also be accompanied by precipitation shafts emanating near cloud base. For example, on 5 April, the maximum occurrence of total clouds is observed at around 900-1000 m (grey), while precipitating-clouds are observed most frequently near 1200 m with secondary peaks near cloud base (black). The same patterns are
shown on 3/23 and 4/7. Figure 14 shows that the second type of precipitation (cloud-top precipitation) is more frequently observed during BACEX. One of the examples of this type of precipitating clouds is shown in Fig. 15 based on photo and radar measurements. Cloud-top precipitation shafts, accompanied by precipitation shafts emanating from the cloud base, are evident from the photo (Fig. 15a). These precipitation shafts are shown with strong radar reflectivity ($Z > -20$ dBz) in Fig. 15b and downdraft (e.g., $V_r < -3$ m s$^{-1}$) in Fig. 15c.

The existence and predominance of the second type of precipitation in shallow marine cumulus could affect the hydrological cycle and cloud radiative forcing. For example, the detrainment moistening and evaporative cooling near cloud top (e.g., Fig. 15) could destabilize the local environment and promote deeper clouds (e.g., preconditioning; Blade and Hartmann, 1993), and further, the deeper and wetter clouds would tend to precipitate more, offsetting the tendency for aerosols to suppress precipitation (e.g., Stevens and Seifert, 2008; Stevens and Feingold, 2009). However, it should be also
noted that aerosol effects on clouds and precipitation are tangled with meteorological influences, which has led to considerable disagreement on the impacts of aerosols on precipitation, both in direction (e.g., decrease or increase) as well as magnitude (Stevens and Feingold, 2009). Lonitz et al. (2015) analyzed non-precipitating clouds that similar to our study and concluded that small changes in the relative humidity can have similar influence on the development of rain as large changes in aerosol concentration, and that aerosol effects on the formation of precipitation are likely very difficult to separate from
co-varying meteorological perturbations. The interactions among clouds, precipitation and aerosols from BACEX will be discussed in a separate study.

        Cloud fields documented during CARRIBA projects also showed similar structures to Fig. 14. For example, frequency distributions showed bimodal peaks in the radar returns, one near cloud base and the other near cloud top (Nuijens et al., 2014). However, the peaks observed near cloud tops in this study were attributed to cloud top precipitation and not to
stratiform clouds (Nuijens et al., 2014) nor to extended cloud layers near cloud tops as a result of stronger inversion (Siebert et al., 2013). Our analysis was confined to shallow marine cumulus clouds (which eliminated the possibility of Sc) and identified precipitating and non-precipitating clouds, which facilitated identification of the peaks near the cloud tops as the precipitation.



The aircraft in situ observations are used to determine how frequently clouds precipitate during the BACEX flights. The daily percentage of precipitating clouds among the total number of clouds observed is shown in Fig. 16. The percentage of precipitating clouds for a given day is estimated by the ratio of precipitating clouds to the total number of clouds sampled. A cloud is counted only if the PVM-100 liquid water content is larger than 0.02 g m$^{-3}$ for more than 3 seconds (~ 180 m

wide). The cloud is classified as precipitating if the precipitation liquid water content, PLWC (derived from the CIP) for a given cloud is larger than 0.1 g m$^{-3}$. However, the choice of the threshold is arbitrary.

The total number of cloud penetrations made on each day ranged from 50 to 200 (not shown). However, the aircraft sometimes penetrated the same cloud more than once, and sometimes avoided clouds with strong updrafts or downdrafts. Nevertheless, Fig. 16a shows that 56 % of the clouds, on average, sampled during BACEX precipitate somewhere in the

10 cloud, and thus about 44 % of clouds are non-precipitating clouds, based on our criteria. This finding is consistent with the percentage of non-precipitating clouds estimated from the radar measurements shown in Fig. 14; no precipitation is observed on 5 of the 12 flights (~ 42 %). In Fig. 16a, the percentage of precipitating clouds at cloud base (grey) shows lower values compared with the percentage of precipitating clouds that were averaged from all level flights (flight-averaged; black). This further confirms that the dominant form of precipitation shafts is not cloud-based precipitation. Although more than about

15 half of the clouds precipitate (Fig. 16a), precipitation rates in and around the cloud during BACEX (Fig. 16b) were far less than 10 mm day$^{-1}$ (2.7 mm day$^{-1}$ on average). Cloud-base precipitation rates on RF02 (3/22), RF08 (3/30) and RF13 (4/7) were larger than those estimated from all the flights (i.e., flight-averaged). By contrast, precipitation rates estimated from all the flights (gray) were larger than those estimated from cloud-base flights for the rest of the days.

## 4 Summary and discussion

In this study, we examined the variations and properties of aerosol, cloud and precipitations over the Eastern Caribbean by using data collected during the Barbados Aerosol Cloud Experiment (BACEX), which took place off the Caribbean island of Barbados from 15 March to 15 April 2010. The marine environment near Barbados provided an excellent area to sample shallow marine clouds with a strong propensity to precipitate. In addition, African dust outbreaks periodically affected the region and provided an excellent opportunity to observe aerosol-cloud-precipitation interactions. The primary observing

platform for the experiment was a Center for Interdisciplinary Remotely Piloted Aircraft Studies (CIRPAS) Twin Otter (TO) research aircraft, which were equipped with standard meteorological instruments, a zenith pointing cloud radar, and probes that measured aerosol, cloud, and precipitation characteristics.

During the one month experiment period, the most intense African dust events during all of 2010 (1-2 April) were observed. Temporal variations and vertical distributions of aerosol observed on the 15 flights, made by the TO research

aircraft, showed a wide range of aerosol conditions. The 10 day back trajectories of air masses observed at Barbados showed three distinct air masses: typical maritime, Saharan, and mid-latitude. These types match well the results from Dunion (2011), who examined about 6000 rawinsonde observations from the Caribbean Sea region taken during the hurricane



seasons, 1995-2002. A variety of aerosol vertical structures were observed and categorized by three distinct profiles associated with aerosol source regions. First, accumulation mode aerosol concentration ($N_a$) decreased with height steadily, with a maximum $N_a$ below the trade-wind inversion near the surface ($< 250$ m g$^{-1}$). These profiles were associated with typical maritime air masses. The second type of profile had $N_a$ increasing with height, with a maximum $N_a$ above the inversion height. These profiles were associated with air masses that originated in the mid-latitudes (east of U.S. and Canada). The third type of aerosol profile was associated with African dust events (31 March - 5 April) where high $N_a$ was observed throughout the entire boundary layer with stratified aerosol structures. Further, this study shows that under some conditions, in which high aerosol concentrations were observed above the inversion but were not transported to the surface, the AOD may not be a good indicator of the boundary layer aerosols that are important to the development of low-level Cu.

Aerosol particle size distributions (PSDs) from dusty days showed a significantly higher $N_a$ for particle diameter between 0.5 µm and 10 µm, compared with PSDs obtained from non-dusty days. The increase of $N_a$ in this particular range may have an important impact on aerosol-cloud-precipitation interactions because the most effective GCCN size lies within this range (Segal et al., 2004; Jung et al., 2015), implying that if the dust particles were coated with hygroscopic material, dust may effectively serve as GCCN (Levin et al., 2005).

Despite the large differences in the total mass loading and the origin of aerosols, the overall shapes of PSDs in the accumulation mode were consistent (except for the two single days of transition occurring before and after a dust event). However, it should also be noted that the slight differences between the individual PSDs could provide some insights into the processes that affect the aerosol concentration via cloud processes (Jung et al., 2013). The observations will be useful for testing how well GCMs can reproduce the aerosol measurements.

During the experiment, the TO research aircraft was able to sample many clouds in various phases of growth. Vertically pointing cloud radar provided the basis for the general characteristics of clouds. Clouds sampled during BACEX had a maximum cloud depth of less than 3 km. However, it is shown that more than half of the clouds precipitate somewhere in the cloud (56 % on average), even though the precipitation amount in and near the cloud was less than 10 mm day$^{-1}$ as a whole (2.7 mm day$^{-1}$ on average). In addition, clouds were far from adiabatic as noted in previous studies (Warner, 1955; Rauber et al., 2007; Gerber et al., 2008), indicating that the adiabatic assumption is not valid in these shallow marine cumulus clouds. Further, clouds thicker than 500-600 m showed a significant chance of precipitating, confirming the early work of the depth required for precipitation in shallow marine cumulus clouds (Malkus, 1958).

Two types of precipitation features were observed during the experiment. In the first type, precipitation shafts emanated mainly from cloud base (i.e., classic cloud-base precipitation) that led to evaporation in the sub-cloud layer. Two, precipitation shafts emanated mainly from near cloud top (or the upper parts of the cloud), with evaporation occurring in the cloud layer. The latter type of precipitating cloud was shallower than the former type, and was also sometimes accompanied by precipitation shafts emanating near cloud base. During BACEX, the cloud-top precipitation type was more frequently observed than the classic cloud-base precipitation type. These two types of precipitation patterns may impact on the trade-wind boundary layer in different ways. For instance, precipitation shafts that emerge from cloud top and evaporate in the



cloud layer (i.e., cloud-top precipitation type), destabilize the atmosphere below the precipitation and provide moisture to the local environment that may affect the moisture budget and lead to an increased cloud lifetime of subsequent clouds (e.g., Albrecht, 1981) and/or later promote deeper clouds (e.g., preconditioning; Blade and Hartmann, 1993). The deeper, wetter clouds would tend to produce more rain, which would offset the tendency for aerosols to suppress rain (e.g., Stevens and

5    Seifert, 2008; Stevens and Feingold, 2009), a topic we will address in a follow-on study.

## 5 Acknowledgements

We thank all individuals who made the observations on the CIRPAS Twin Otter during BACEX. We thank Dr. Joseph M. Prospero (U. of Miami) for providing dust surface data and he and his staff for establishing and maintaining the Ragged Point AERONET sites used in this investigation. Jung and Albrecht are funded by ONR Grant N000140810465. Feingold

10   acknowledges support from NOAA's Climate Goal. EJ thanks Robert Seigel (publiscize.com) for scrutinizing an early stage of the manuscript.




# Appendices

**Table A1.** Flight list.

| Flight | Date | Time (UTC*) | Number of soundings | Note |
|---|---|---|---|---|
| RF01 | 19 Mar | 15:14 – 16:40 | 2(2) | Spuriously high CAS $N_d$ on this flight |
| RF02 | 22 Mar | 15:01 – 16:28 | 2(2) | - |
| RF03 | 23 Mar | 14:28 – 18:20 | 4(3) | No PCASP data available, cloud chaff, clear air chaff |
| RF04 | 24 Mar | 14:50 – 18:29 | 4(2) | Cloud chaff, clear air chaff |
| RF05 | 25 Mar | 14:50 – 17:52 | 4(2) | Clear air chaff |
| RF06 | 26 Mar | 14:45 – 16:04 | 2(2) | Cloud chaff |
| RF07 | 29 Mar | 15:03 – 19:02 | 4(2) | Two Clouds chaff, clear air chaff |
| RF08 | 30 Mar | 14:40 – 18:12 | 6(2) | Strong downdraft from cloud outflow |
| RF09 | 31 Mar | 14:40 – 18:10 | 4(2) | African dust outbreak |
| RF10 | 1 Apr | 15:14 – 18:18 | 3(2) | African dust outbreak, no cloud |
| RF11 | 2 Apr | 14:40 – 17:18 | 3(2) | African dust outbreak, few clouds, clear air chaff |
| RF12 | 5 Apr | 14:48 – 18:29 | 1(1) | African dust outbreak, water spout observed |
| RF13 | 7 Apr | 14:33 – 17:17 | 3(2) | - |
| RF14 | 10 Apr | 14:36 – 17:19 | 3(2) | - |
| RF15 | 11 Apr | 14:36 – 18:09 | 3(2) | No CCN data available |

*Local time: UTC-5

* The total number of soundings is included take-off and landing soundings. The number of the $n^{th}$ sounding used in Fig. 2 is

5    shown inside parenthesis.



**Table A2**. Table of acronyms and symbols.

| Acronym | Expression |
| --- | --- |
| AERONET | Aerosol robotic network |
| AOD | Aerosol Optical Depth |
| BACEX | Barbados Aerosol Cloud Experiment |
| CAS | Cloud Aerosol Spectrometer |
| CCN | Cloud Condensation nuclei |
| GCCN | Giant CCN |
| CIP | Cloud Imaging Probe |
| $D_a$ | Aerosol particle size |
| $D_e$ | Cloud droplet size |
| HYSPLIT | Hybrid Single Particle Lagrangian Integrated Trajectory |
| LCL | Lifting condensation level |
| LWC | Liquid water content |
| $N_a$ | Aerosol number concentration |
| $N_d$ | Cloud droplets number concentration |
| PCASP | Passive Cavity Aerosol Spectrometer Probe |
| PSD | Particle Size Distribution (used for aerosol particles) |
| SAL | Saharan Air Layer |
| TO | Twin Otter (research aircraft) |





**Table A3**. The lists of various vertical structures and air masses during BACEX.

| RF # | Date | Origin of air mass (Fig. 4) | Vertical structure (Fig. 5) | Note |
|---|---|---|---|---|
| RF01 | 19 March | Africa (SAL) | - | - |
| RF02 | 22 March | Ocean | Type A | - |
| RF03 | 23 March | MLDA | - | - |
| RF04 | 24 March | MLDA | - | - |
| RF05 | 25 March | MLDA+Ocean | Type B | - |
| RF06 | 26 March | MLDA+Ocean | Type B | - |
| RF07 | 29 March | Ocean | Type A | Pre-dust |
| RF08 | 30 March | Transition | Type A | - |
| RF09 | 31 March | SAL | Type C | Dust period |
| RF10 | 1 April | SAL | Type C | Dust period |
| RF11 | 2 April | SAL | Type C | Dust period |
| RF12 | 5 April | SAL | Type C | Dust period |
| RF13 | 7 April | Transition | Type A | - |
| RF14 | 10 April | Ocean | Type A | Post-dust |
| RF15 | 11 April | Ocean | Type A | Post-dust |

*Type A: aerosol concentrations decrease with height monotonically.

*Type B: high aerosol concentrations confine above trade-wind inversion

*Type C: high aerosol concentrations prevail throughout the boundary layer and/or complicated structure.

5    *MLDA: Middle Latitude Dry Air,  SAL: Saharan Air Layer





**Table A4.** Cloud-base level-run flights for the radar analysis.

| RF # | Date | Time (UTC*) | Flight height | Note |
|------|------|-------------|---------------|------|
| RF01 | 19 Mar | - | - | - |
| RF02 | 22 Mar | 15:52:48-16:10:48 | 1035 m | Heavily precipitating cloud |
| RF03 | 23 Mar | 17:04:48-17:24:36 | 1065 m | - |
| RF04 | 24 Mar | 17:01:48-17:24:36 | 525 m | Precipitating cloud |
| RF05 | 25 Mar | 15:31:12-16:03:00 | 795 m | Non-precipitating cloud |
| RF06 | 26 Mar | 15:27:54-15:36:00 | 1005 m | Non-precipitating cloud |
| RF07 | 29 Mar | 17:06:00-17:18:36 | 885 m | Non-precipitating cloud |
| RF08 | 30 Mar | 17:36:00-17:49:48 | 405 m | Precipitating cloud, sub-cloud leg |
| RF09 | 31 Mar | 16:53:24-17"07:48 | 705 m | Non-precipitating cloud |
| RF10 | 1 Apr | - | - | No cloud |
| RF11 | 2 Apr | - | - | No decent cloud |
| RF12 | 5 Apr | 16:09:00-16:24:36 | 825 m | - |
| RF13 | 7 Apr | 16:32:46-16:43:12 | 735 m | - |
| RF14 | 10 Apr | 16:19:30-16:28:12 | 1005 m | - |
| RF15 | 11 Apr | 16:02:24-16:21:00 | 795 m | Non-precipitating cloud |

*Local time: UTC-5





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



**Tables**

**Table 1.** Characteristics of instruments used in Barbados Aerosol Cloud Experiment

| Instrument | Observations/Purpose |
|---|---|
| Standard meteorological instruments | Winds, temperature, dew-point, cloud liquid water content, surface temperature, etc |
| Gerber LWC sensor (PVM-100) | Liquid water content (g m$^{-3}$) |
| 95 GHz Frequency Modulated Continuous Wave (FMCW) Doppler radar (zenith viewing mode) | Doppler spectra (Reflectivity, Doppler velocity and spectrum width); Cloud properties, in-cloud turbulence |
| CPCs | Total and ultrafine aerosol, cutoffs at D=3 nm, 10 nm and 15 nm. |
| Passive Cavity Aerosol Spectrometer Probe (PCASP) | Aerosol 0.1 – 2.5 μm, 20 bins |
| Cloud Aerosol Spectrometer (CAS) | Aerosol and Clouds 0.6 – 60 μm, 20 bins |
| Cloud Imaging Probe (CIP) | Drizzle 25 – 1550 μm, 62 bins |
| CCN-200 | CCN (super-saturation at 0.3 %, 0.6 %) |



**Figures**



**Fig. 1.** Time-height cross-section of (a) relative humidity (%) (b) potential temperature (K)  (c) wind speed (m s$^{-1}$) and (d) wind direction (degrees), obtained from soundings launched from Barbados at 12:00 UTC from 14 March to 16 April 2010. Days of the first (19 March) and the last (11 April) flights are denoted as solid, black vertical lines. A period of heavy African dust (31 March-5 April) is denoted by the dashed, black vertical lines. The primary and secondary inversion heights are shown as square and cross symbols, respectively. Lifting Condensation Level (LCL) and 0 ℃ isotherm are overlaid in Fig. 1(a) and (b) as black lines connected with circular symbols. The LCL is calculated by lifting a parcel with the average thermodynamic properties for the layer 100 – 200 m above the ocean surface. Sounding data were obtained from the University of Wyoming's online Upper Air Data (http://weather.uwyo.edu/upperair/sounding.html).





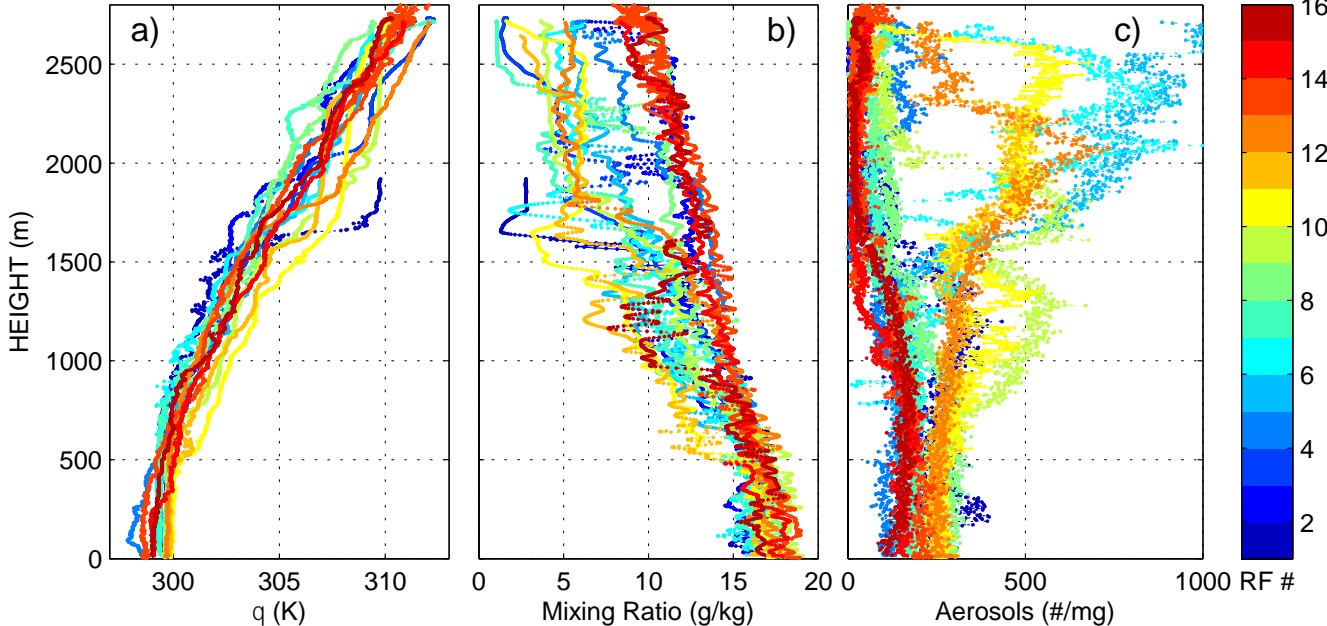

**Fig. 2.** Profiles of (a) potential temperature, Θ, (b) water vapor mixing ratio (g/kg), and (c) aerosol number concentration per mass of air (#/mg) obtained from PCASP during the aircraft's ascents and/or descents. The profiles shown are one out of many soundings for each day and are denoted in Table A1. The color bar shows the number of research flight (RF #), shown in Table A1.





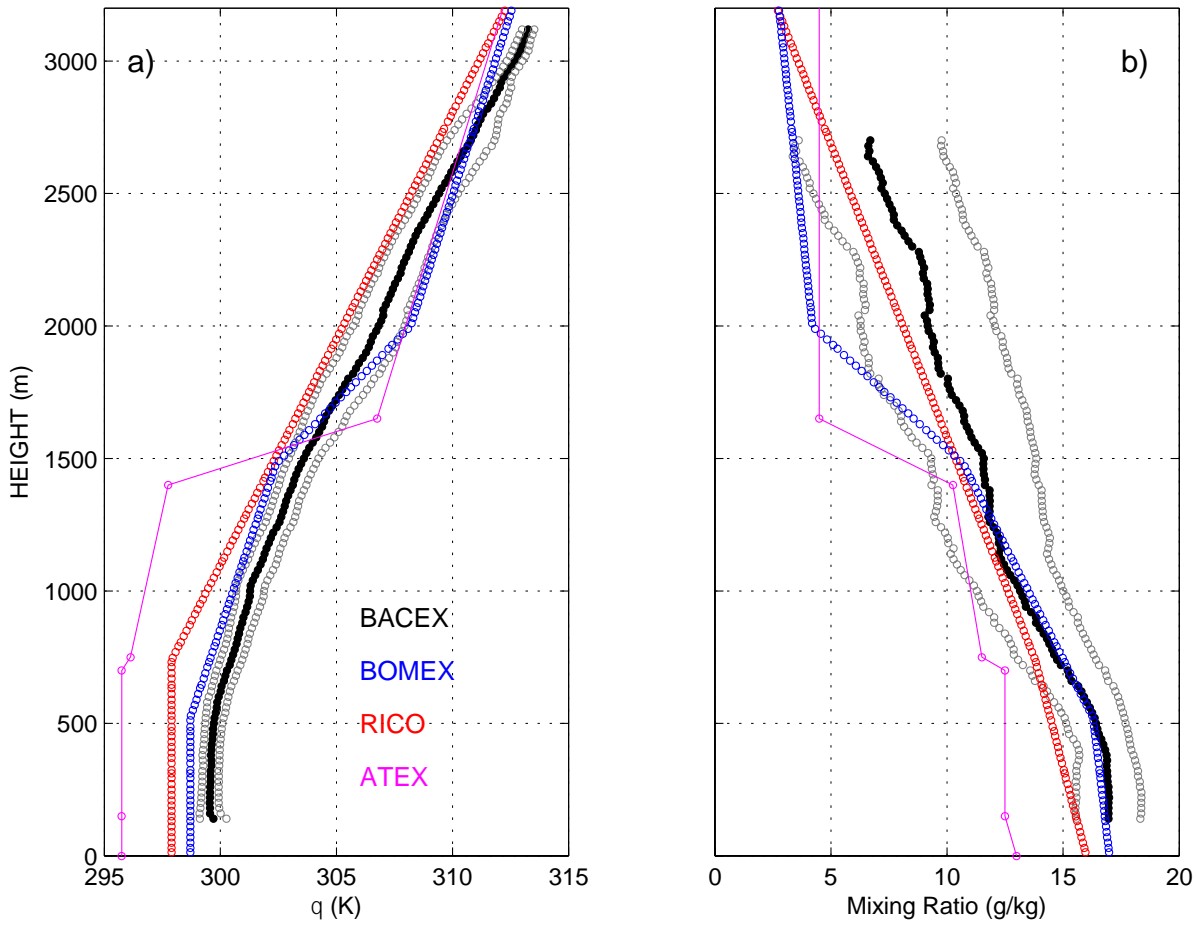

**Fig. 3.** Profiles of (a) potential temperature, Θ, water vapor mixing ratio obtained from BACEX (black) with ±1σ (grey), BOMEX (blue), RICO (red) and ATEX (magenta) field campaigns. Data of BOMEX, RICO, and ATEX are obtained from GCSS (GEWEX Cloud System Study) boundary layer cloud homepage. BACEX profiles are obtained from all data sampled during the experiment.




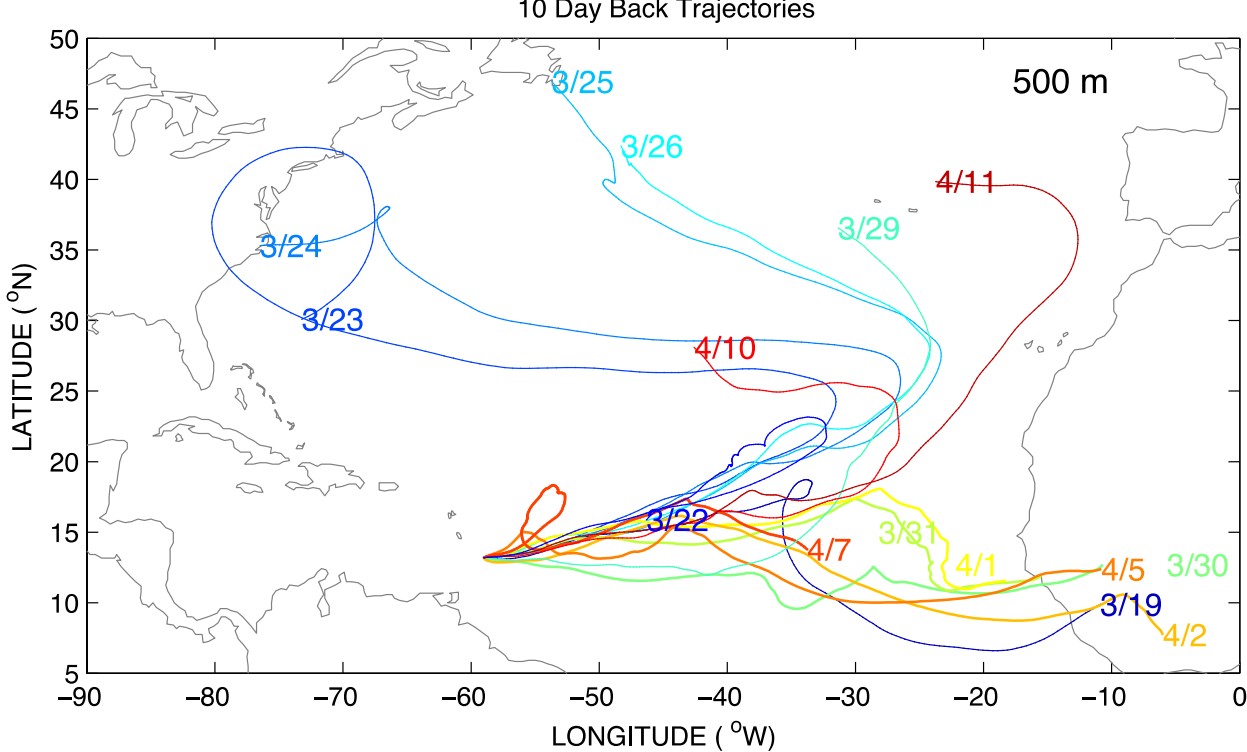

**Fig. 4.** The 10 day back trajectories, arriving at 500 m in the middle of the BACEX flight domain. Dates for each back-trajectory are shown accordingly.





**Fig. 5.** (a) Vertical distribution of the accumulation mode aerosol (PCASP) obtained from aircraft ascents or descents where aerosol concentration is offset by 400 mg⁻¹ for each flight. CCN (super-saturation = 0.6 %) are plotted on 23 March for vertical profiles since no PCASP is available on this day, (b) temporal variation of aerosol at sub-cloud layer during BACEX, and (c) Dust concentration recorded at the Barbados Ragged Point surface site (13.2 °N, 59.5 °W). Level 2 Aerosol Optical Depth (AOD) at 500 nm wavelength (red) from AERONET is shown. Dust data are provided by Dr. Joseph M. Prospero of the University of Miami.





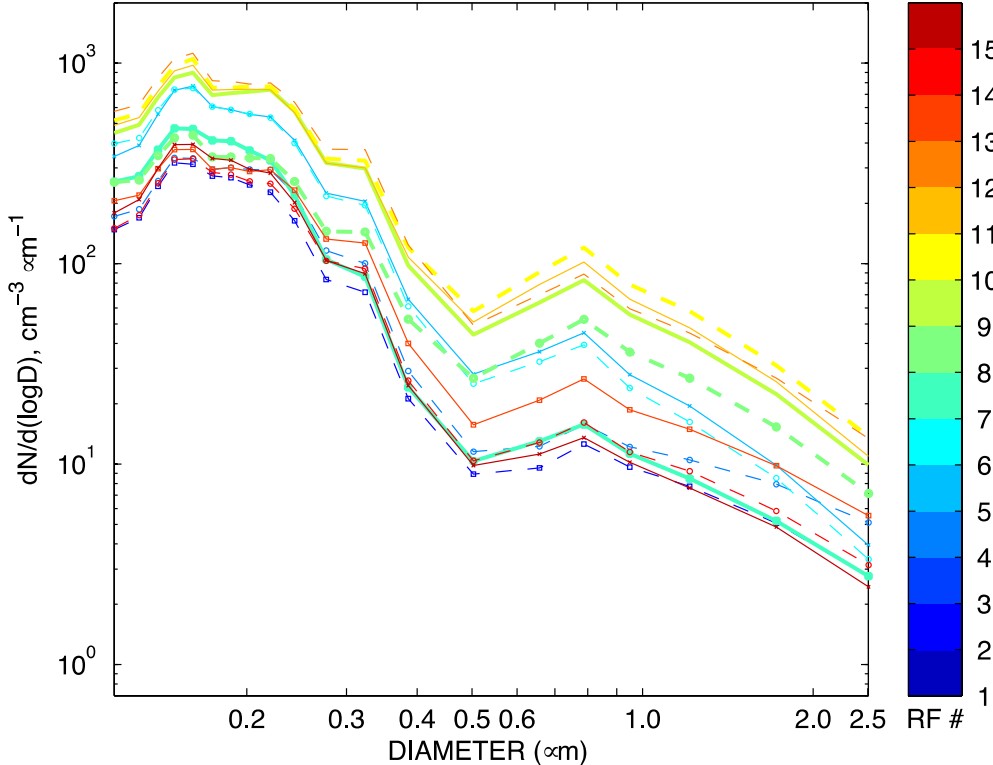

**Fig. 6.** Daily averaged aerosol particle size distributions (PSDs) ranging from 0.1 µm to 2.5 µm obtained from the PCASP. Color bar indicates the research flight number (RF #), shown in Table A1. PSDs from the odd (even) RF numbers are shown as solid (dashed) lines. PSDs estimated between RF07 and RF10 (3/29, 3/30, 3/31, 4/1) are denoted as bold lines. PSDs of RF01 (3/19) and RF03 (3/23) are not shown due to the instrument malfunction (RF01) and the absence of PCASP data (RF03) for the days.



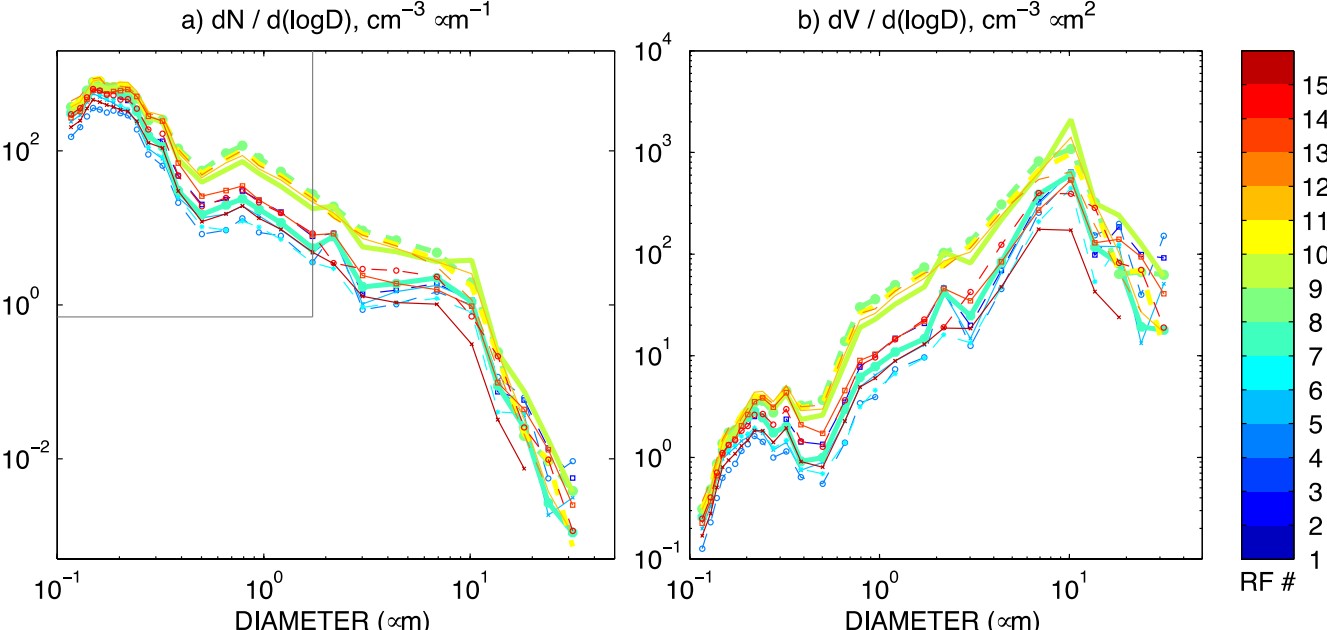

**Fig. 7.** Daily averaged aerosol particle size distributions (PSDs) for the sub-cloud level flights. PSDs obtained from PCASP and CAS probes are combined to obtain PSDs ranging from 0.1 µm to 30 µm. The color bar indicates the research flight number (RF #) presented in Table A1. PSDs from the odd (even) number of RF are shown as solid (dashed) lines. PSDs between RF07 (29 March) and RF10 (1 April) are denoted as bold lines. PSD of RF01 (19 March) and RF03 (23 March) are not shown due to the instrument malfunction and the absence of PCASP data for the days. The scale of Fig. 6 is shown as a box in Fig. 7a in the upper-left corner.



**Fig. 8.** Time-height cross section of reflectivity on (a) 22 March, (b) 24 March, (c) 29 March and (d) 11 April, 2010 from the cloud-base level flight during 5-minute periods (about 18 km in horizontal extent) at an air speed of about 60 m s$^{-1}$. Data were sampled from (a-b) precipitating and (c-d) non-precipitating clouds.



**Fig. 9.** MODIS satellite images on (a) 22 March (b) 24 March (c) 29 March and (d) 30 March 2010 for area near Barbados. The flight domains are shown as red dotted boxes. The outer box indicates the average flight domain during BACEX. The flight domain of the particular day is overlaid as an inner box if the satellite image is obtained during flight periods. The numerical number shown at the lower- right side of the figure indicates Julian day in UTC (e.g., 088.1730 indicates Julian day 088, 1730 UTC). Images were obtained from the MODIS website (http://modis-atmos.gsfc.nasa.gov/IMAGES/index.html).




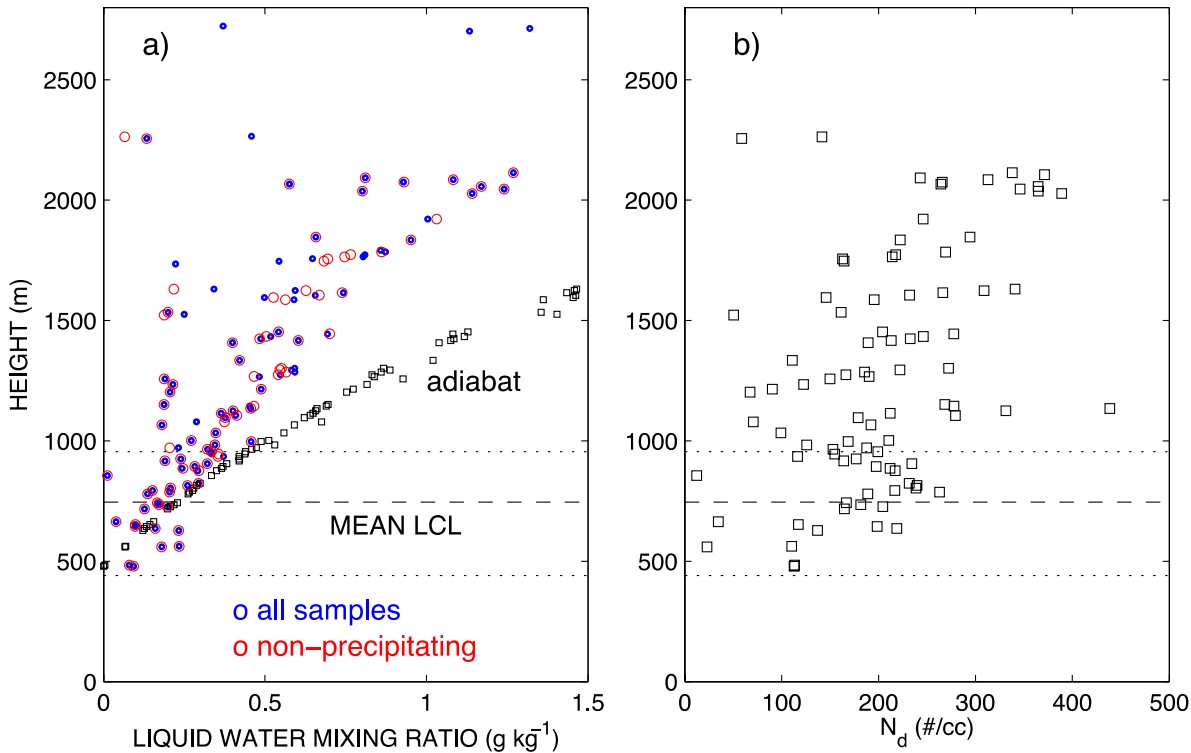

**Fig. 10.** (a) Cloud water and (b) droplet number concentration $N_d$ in cloud core ($w > 1$ m s$^{-1}$) sampled by the Twin Otter during BACEX. Non-precipitating samples (CIP volume < 0.01) are used to estimate $N_d$ in Fig. 10b. Mean, minimum and maximum values of LCL are denoted by dashed, and dotted lines.





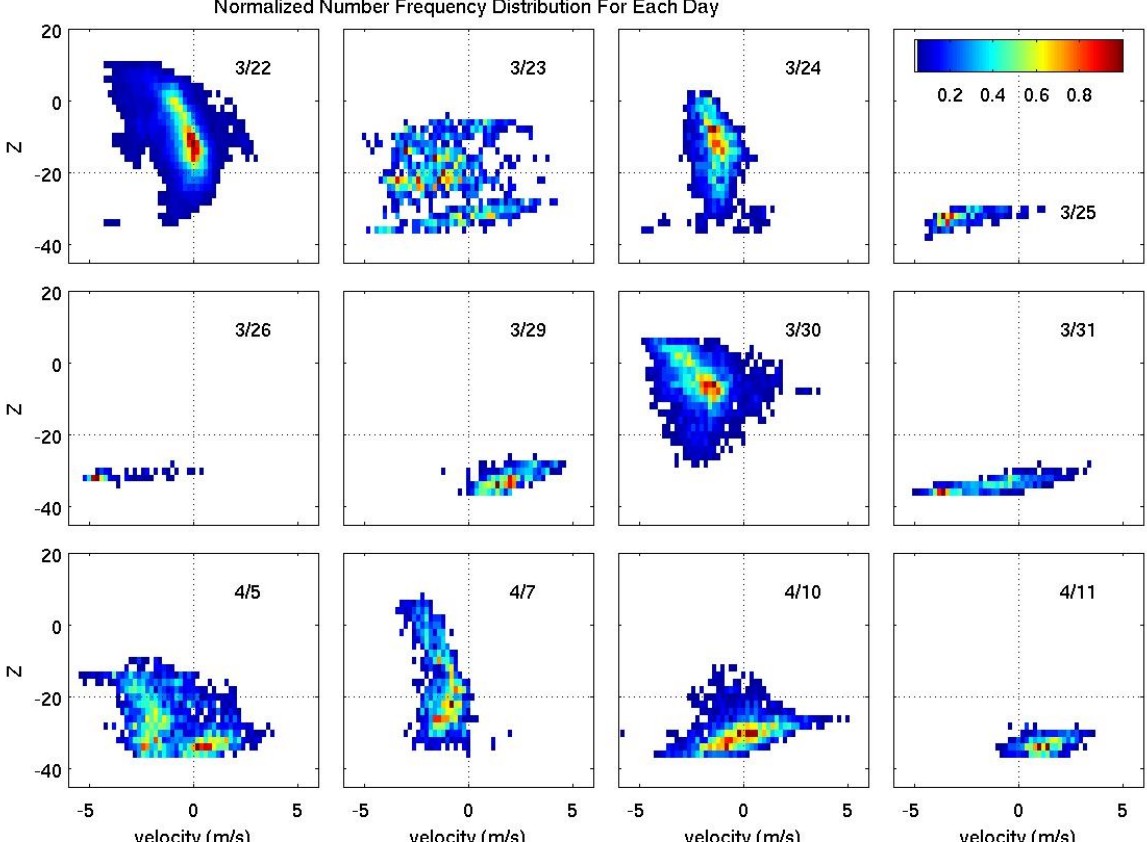

**Fig. 11.** Normalized velocity-reflectivity number frequency distributions on each day during BACEX from the cloud-base level-leg flights. Intervals of 2 dBz, and 0.1 ms$^{-1}$ are used to obtain the frequency distribution. Positive Doppler velocity indicates an upward motion. The color bar is displayed in the upper right corner. The reflectivity of -20 dBz and Doppler velocity of 0 m s$^{-1}$ are denoted by the dotted line. No clouds were observed on 1-2 April during the cloud-base level flights. The time and periods of each cloud-base level flight are listed in Table A4.





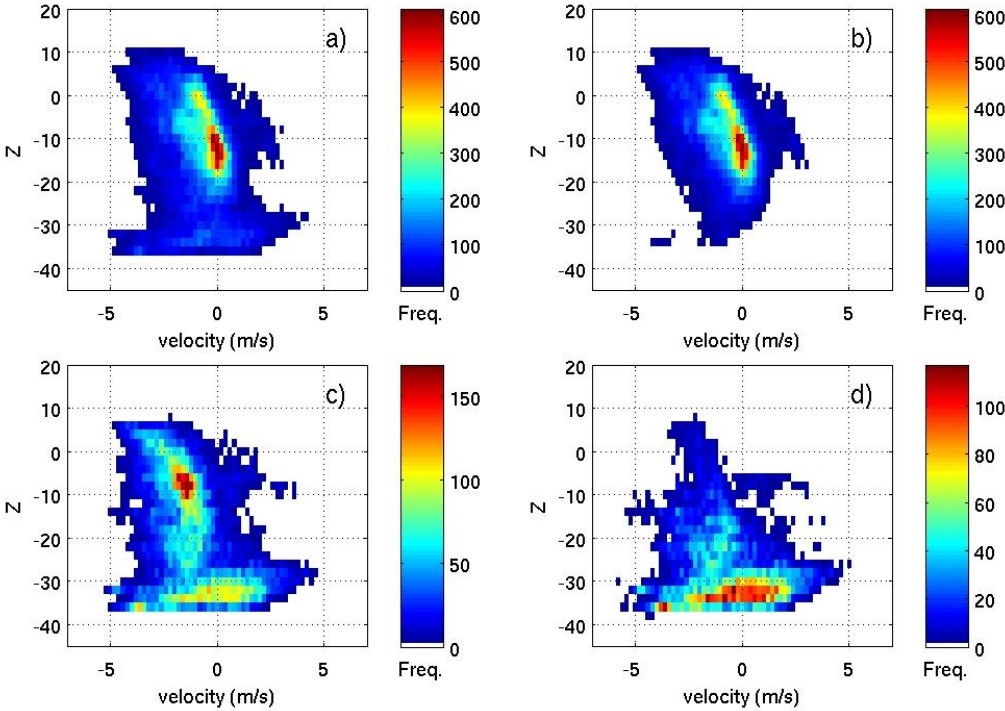

**Fig. 12.** Cloud reflectivity and velocity distributions estimated from an average of all individual days (12 cases in Fig. 11), (b) using three precipitating clouds days (clouds sampled on 3/22, 3/24, and 3/30) and (c) using 11 days except for clouds on 22 March, which sampled the strongest precipitating clouds, and (d) from non-precipitating and/or lightly precipitating clouds (remaining 9 days in Fig. 11).





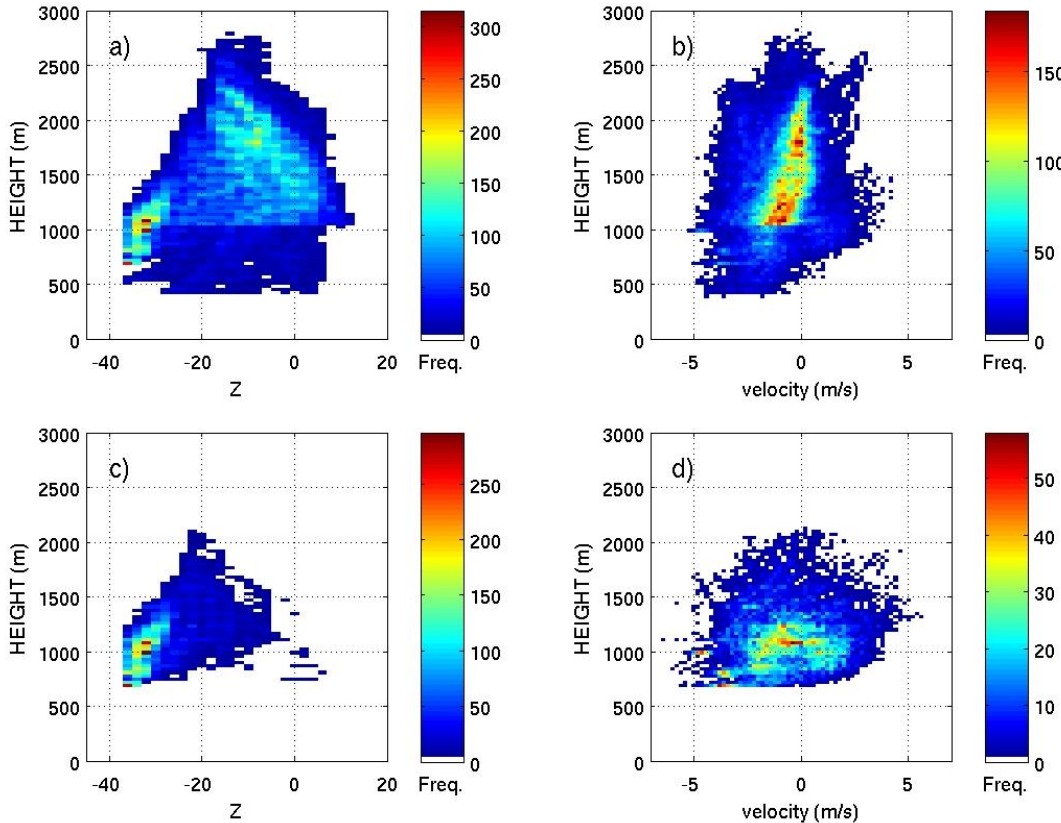

**Fig. 13.** Frequency distribution of reflectivity and velocity with heights, by compositing (a-b) all available days (12 days in Fig. 11), and (c-d) the nine days excluding the major precipitating clouds sampled on 22, 24, and 30 March 2010. Intervals of 30 m (vertical), 2 dBz, and 0.1 m $^{-1}$ are used to obtain the frequency distribution.





**Fig. 14.** A normalized number of samples with heights for all sampled clouds (grey) and precipitating clouds (black). Precipitating clouds are defined as data points with Z > -20 dBz and vertical velocity < 0 m s$^{-1}$. No precipitating clouds are observed on 25, 26, 29, 31 March and 11 April.





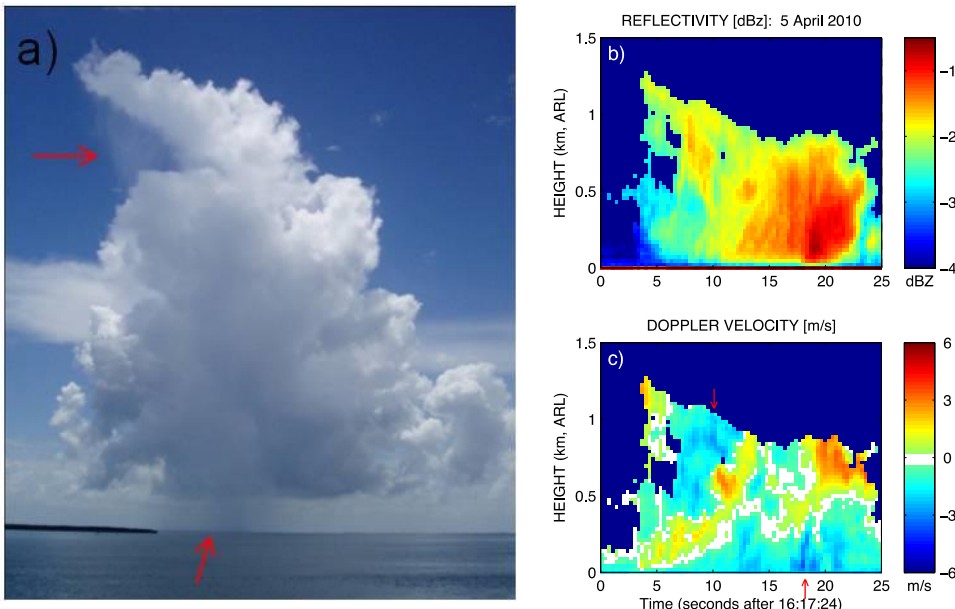

**Fig. 15.** (a) A cloud photo, radar (b) reflectivity and (c) Doppler velocity that show two types of precipitation; Precipitation shafts emanate near the cloud top on the downshear side of the cloud; Precipitation shafts emanate near the cloud base. (The photo is of a cloud over Key Biscayne in an environment similar to that in Barbados). The radar returns in Fig. 15(b-c) are obtained from a cloud sampled on 5 April 2010 during BACEX.



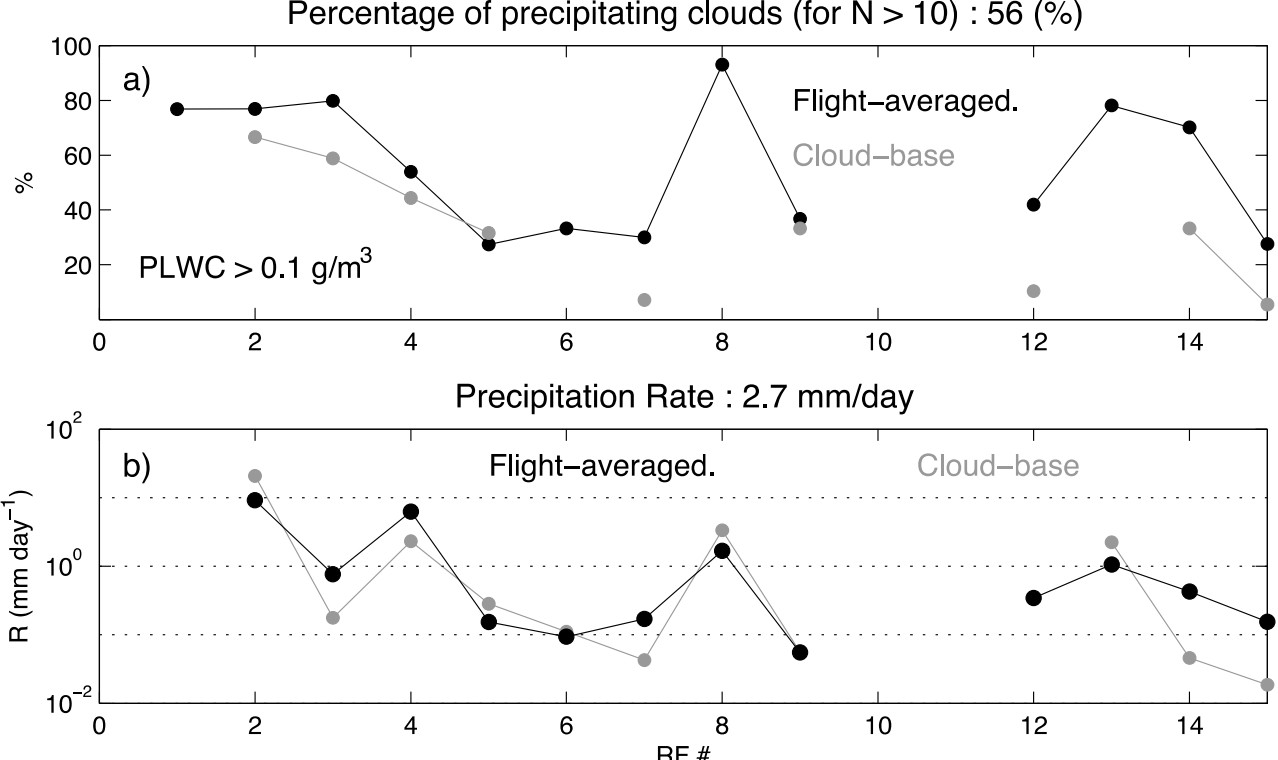

**Fig. 16.** (a) Percentage of precipitating clouds estimated from all clouds sampled (black; flight-averaged), and from clouds sampled during the cloud-base flights (grey; cloud-base) with a threshold of precipitation liquid water content (PLWC) larger than 0.1 g m$^{-3}$. The CIP probe volume concentration (cm$^3$ m$^{-3}$) is multiplied by the density of water to obtain PLWC.

5   (b) Flight-averaged (black) and cloud-base (grey) precipitation rate (mm day$^{-1}$).