# Peer review of "Aerosols, Clouds, and Precipitation in the North-Atlantic Trades Observed During the Barbados Aerosol Cloud"

_Atmospheric Chemistry and Physics, 2016_

## Referee Comment (RC1) · Anonymous Referee #1 · 28 Mar 2016

Review on *"Aerosols, Clouds, and Precipitation in the North-Atlantic Trades Observed During the Barbados Aerosol Cloud Experiment. Part I: Distributions and Variability"* by Jung et at.

**General Comment:**

This paper presents a nice overview of „aerosol, cloud, and precipitation features" as measured over Barbados. Airborne data has been sampled in-situ combined with a cloud radar during 12 research flight. My overall opinion is that this paper has the clear motivation to characterize the observations without going into much detail of individual findings. I have no concerns about this strategy, in particular because the observation period nicely covers the three typical aerosol types typically observed at Barbados. However, at some points (see specific comments) a somewhat deeper analysis and discussion of individual findings would improve the paper instead of refering to another promised upcoming paper.

However, I have a few major critical points which have to be seriously discussed before I can suggest this paper for publication.

1.) **Page 5, beginning of sec 2.5:** I have serious doubts about this method of simply combining two size distributions with one of it is sampled under ambeient and the other under dry conditions. The two size distributions may line up well but this could be really by chance! A careful analysis of this issue is absolute necessary. A lot of bigger aerosol might be sea salt particles which are highly hygroscopy and there should be a big difference if you measure it under dry or ambient conditions. Please include at least error bars which describe this effect! This is a serious point which needs more careful discussion.

2.) **Page 9, Sec 3.3.2:** Do I correctly understand? The PSD is averaged over all flight height for one individual flight - including sub-cloud layer and cloud layer heights? Does this make sense? Why not at least distinguish between sub-cloud layer and cloud layer aerosol? I have serious doubts about the representativiness of these size distributions. In particular for the situation where dust is advected the size distribution should be a strong function of height.

Furthermore, all size distributions show more or less exactly the same general structure/shape with characteristic peaks and shoulder at the same size. It looks like that the distributions differ only due to dilution although you mentioed that you had three different types of aerosol loading? Although the y-axis is logarithmic this seems quite strange. I discussed this issue with an expert for OPC measurements and we both wonder if this could be an instrumet artefact. Please discuss this issue in detail.

3.) **Page 16 (Summary):** This section is quite often a word-by-word summarize of the previous sections. The discussion is missed out completely. I totally understand that for a data overview paper a discussion is not a trivial task but repeating all the obersations word by word is not a convincing solution.

**Specific Comments:**

Page 2, line 3ff: How are your findings biased by the sampling strategy - is it natural to assume that one tries to hit the bigger clouds which bias the frequency of observed cloud types?

P2, l 15ff: One might mention at least a few open questions. "Cloud have to better understood" is a quite generic statement.

P2, l 17: Don't forget Malkus' landmark papers here; although these experiments were smaller three nice papers came out o fit.

P2, l 31: Why is Barbados a perfect place for such studies?- Be more specific here.

P 4, l 6: "PVM-100 water content" is not a parameter – please use liquid water content instead.

P4, l 8: Provide location of the company, use „Inc" instead „inc".

P4, l 10ff: You should mention at this point which instrument measures under environmental conditions and which one is a closed system which samples under dried conditions – it's important here!

P4, title of sec 2.3: Better "stratification"? Large-scale would also imply a horizontal component – right?

P4, l 26: Better characterized/analysed instead of „defined" ?

P5, l 10ff: I have serious doubts about this point: The two size distributions may line up well but this could be really by chance! A careful analysis of this issue is absolute necessary. A lot of bigger aerosol might be sea salt particles which are highly hygroscopy and there should be a big difference if you measure it under dry or ambient conditions. Please include at least error bars which describe this effect! This is a serious point which needs more careful discussion (see general comment).

P 6, l 6: You could include the equation (in text) for the LCL determination but not realy necessary

P 6, l 33: Is there a difference/bias between the radiosondes and TO-profiles for determing the inversion? You mentioned earlier the method how to determine the main inversion and here you say that the inversion was poorly defined? Please specify!

P8 , l22: In the plot you abbreviate super-saturation with "ss", here with „s" – be consistent.

P9, l 15: Do I correctly understand? The PSD is averaged over all flight heights for one individual flight – including sub-cloud layer and cloud layer heights? Does this averaging makes sense? Why not at least distinguish between sub-cloud layer and cloud layer aerosol? I have serious doubts about the representativiness of these size distributions. In particular for the situation where dust is advected the size distribution should be a function of height.

Furthermore, all size distributions show more or less exactly the same general structure with characteristic peaks and shoulder at the same size. It looks like that the distributions differ only due to dilution although you mentioed that you had three different types of aerosol loading? Although the y-axis is logarithmic this seems quite strange. I discussed this issue with an expert for OPC and we both wonder if this could be an instrumet artefact. This pont has to be discussed in detail (see general point).

P9, l 18: I do not agree with the statement that PSDs avageraged over sub-cloud and cloud

layer can provide any insight in processes. You average over regions where different processes are dominant. I suggest re-wording.

P 9, l 32: See my comment about combining an aerosol size dristribution measured with a closed system (dried aerosol) and an open syestem (ambient conditions).

P 10, l 27: I don't understand exactly what you mean with "precipitating clouds exhibit more organized mesoscale features"? Can you please specify!

P 11, l 6: About LWC measureemnts with the PVM-100A; it is known that the transfer function oft he PVM-100A decreases with increasing droplet size (see Wendisch, Garrett, and Strapp: 2002). The droplets in trade wind cumuli have comparable big droplets due to the low number concentration and this underestimation of the PVM might be an issue. Although you might not have the droplet sizes available you should at least mention this issue because it could partly result in the bigger discrepancy bebetween measured and adiabatic value around cloud top region. Of course you could take the effective radius derived from the PVM-100A and estimate the transfer function based on this parameter – but maybe mentioning this problem is also fine.

P 11, l 9: I don't completely understand this criterium, does it really completely rule out the possibility of counting shattered droplets? Please explain in more detail.

Furthermore, a few line below you mention CARRIBA_dry: In the "polluted" case of CARRIBA_dry (April 24th) the aerosol number concentration in the SCL was about 265 particles per cm^3 resulting in mean droplet number concentration of 80 droplets per cm^3 (measured with a PDI). In your plots you have aerosol number concentrations of 100 to 300 per cm3 in the SCL (see Fig 2c) (assuming air density about 1 kg/m^3 in the SCL) but resulting in 100 to 200 droplets per cm^3 between z = 500 and 1000 m) in Fig 10b. This means that about 75% oft he aerosol particle will be activated? Furthermore, your profile of N_d is at least slightly increasing with height! It is not very typical that droplet number concentration is increasing with height although processes such as secondary activation might play a role for a few cases. Please discuss this point and offer an explanation for this behaviour.

On a first glance this looks like an inconsistency to me and should be discussed in much more detail. For example, you could show a profile of N_droplets for one flight which can be directly compared to the aerosol profile of the SCL? Then you can estimate how many aerosol particles are really activated. Your data suggest that more than 70% (or so) of the aeroosl as measured in the SCL will be activated

P 11, l 9: Again, the parameter is "LWC" and not "PVM-100 LWC", by the way, in the plots the liquid water mixing ratio plotted and not the LWC.

P 11, l 10: Wasn't there a Phase Doppler Interferometer (PICT) installed? If yes and if it worked properly, you should have a reliable estimate of the droplet number concentration (and size distribution).

P 11, l 15: I don't understand the "comparison" with the CARRIBA_dry campaign in this context. The three differernt types of aerosol origin were also observed during CARRIBA.

End of p 11: I am not an expert in radar meteorology but a liitle bit more detailed discussion would be appreciated. What can I learn from the radar measurements?

P 12, l 6: From reading this part of the text it is difficult to understand the differences of the four patterns in Fig 12. – in particular between panel b) and c). How is "strongest precipitationd" defined and why do you exclude a few precipitating clouds in panel d) ? – On a first glance this seems to be a little bit arbitrary and the motivation for doing this remains unclear. A few more word would help the reader.

P 12, l 22: Have you really excluded three precipitating clouds or three days with precip?

P12, l 24: I am confused here: a few lines above you claimed that clouds deeper 500 to 600 m have a significant chance of precipitating and now you analyze the non-precipitating clouds in panel c) and d) with a thickness of about 1300 m? isn't this contradicitionary or do I misunderstood something? Please specify!

P 13, l 10: One should also mention in the text thtat the photo in Fig 15 is not from BACEX

P 13, l 15: The following part is more discussion and not data analysis. However, I do not see how this discussion is justified by the observed data and its analysis. This part is a little bit confusing to me and should be re-worded.

P 13, l 27: Although there is a strong collaboration between the CARRIBA community and the activity ot the MPI group, the work of L Nuijens et al. should not be considered as part of CARRIBA.

P 14, l 1: I think the frequency of precip should be mentioned earlier – it is a quite fundamental point and comes surprisingly late in this section.

P 14, l 4: How is the precipitation LWC defined here?

About „Summary and Discussion" Overall comment:  This section is really mostly a word-by-word summarize of the previous observations. The discussion part is missed out completely. I totally understand that for a data overview paper a discussion is not a trivial task but repeating all the obersations word by word is not a convincing solution and I suggest a complete re-wording of this section.

P 14, l 32: It is a little bit strange to talk about hurricane seasons for Barbados because Barbados wasn't influenced by a hurricane for several decades (before your flights, the first hurricane „Tomas" came in just a few month later...

P 15, l 2ff: This sentence is quite confusing because one can interpret it such as the maximum of $N_a$ is below the inversion which of course makes no sense if it is steadily decreasing with height – I suggest re-wording to make it clear what you mean.

P 15, l 12: Just a comment but there are very contradicionary studies about a possible influence of GCCN on the development of precipitation – could be at least mentioned here.

P 15, l 16: What do you mean with "could provide" I think at this point a deeper data analysis woud be interesting.

P 15, l 24: I think a little bit more quantitative analysis would be justified at this point! That cumulus clouds are far away from being adiabatic is well known.

Figures:

Fig 1: to include the units in brackets is confusing because one would expect the paramater which was mentioned before - the potential temperature is "\Theta" and not "K" – just a formal point.

Fig 1: Although I would do it in the same way: is there a good argument for taking the readings in the height range between 100 and 200 m? I assume the radiosonde takes some time to provide  reliable readings? If you would take data from the TO you could take the 30m level legs - right?

Fig 2: Although it is a nice overview details of the profiles are difficult to detect - this is particular true for \Theta and q_l.

Fig 2c: The unit „particles per mg" - a kind of aerosol number mixing ratio - is not very common (although it makes sense...). Usually I would expect particles per cm^3 normalized to standard conditions - please specify in the text what you did here.

Fig 2a: In my pdf copy the "\Theta" does not show up.

Fig 2 caption: Why do you use sometimes symbols, sometimes not? I suggest to use symbols in the xlabels of the plot and in the caption I suggest using "water vapor mixing ratio $q_l$ (g/kg)" and being consistent with this notation throughout the paper.

Fig 3: You could consider plotting the 1-sigma as classical error bars or at least dotted lines. First I was confused because it looks like another data set but you have described it in the figure caption so everything is technicaly correct.

Fig 5: Difficult to figure out the absolute values.

Fig 5, ylabel: Is here a "micron" missing? it does not show up in my pdf copy? Please check all figures for this problem.

Fig 9: The red dashed lines should be a little bit thicker!

Fig 10: From this profiles it looks like that cloud base is 250 m below LCL ? and around cloud base you have super-adiabatic $q_l$?

I don't understand the calculation of adiabatic $q_l$; is it estimated for each cloud core individually? If so, why do you have super adiabatic $q_l$ between mean LCL and minimum LCL? Why not normalizing each cloud with its adiabatic value or showing the difference between actual $q_l$ and adiabatic value for each cloud?

---

## Referee Comment (RC2) · Anonymous Referee #2 · 29 Apr 2016

Review of: Aerosols, Clouds, and Precipitation in the North-Atlantic Trades Observed During the Barbados Aerosol Cloud Experiment. Part I: Distributions and Variability By Jung, Albrecht, Feingold, Jonsson, Chuang and Donaher.

Evaluation: Publish with minor revisions

Major comments:

This very much reads as a paper setting the stage for something to come. As such it is a bit light, but it does contain sufficient information to warrant publication.

Minor comments:

Page 5 line 11: "The PCASP dries the particles before measuring them." Please provide a reference for this. For instance, Strapp et al (1992, JAOTech) leaves the door open for the PCASP only partially drying particles larger than sub-micrometer (their summary). Thus the behavior may be quite different for a dust particle (or for a coated dust particle) and a sea-salt particle. Other references?

Page 5 line 15: Missing integration sign in denominator.

Page 9 lines 24-26: I am not sure what you are implying here; does sea-salt not contribute to the larger sizes in Fig. 6? This would seem to be inconsistent with many studies that have found sea-salt in the entire range of marine aerosols, e.g. Clarke et al. (2006, JGR), Blot et al (2013, JGR), Modini et al (2015, JGR).

Page 10, lines 8-9: A reference for the sizing uncertainty?

Page 31 and rest of manuscript: Are you connecting a PCASP (which you claims dries particles), with a CIP which does not materially dry dust particles (except maybe for a thin coating layer) and which when measuring sea-salt will see un-dried hydrated particles. I do not see any discussion of the fact that you do not necessarily know what is being looked at with the CIP; maybe I missed it?

Page 11 line 9: What is a "CIP volume number concentration"? It occurs several times.

Page 11 line 20: "increases with height." This is one of the places that the manuscript comes up missing some context. No discussion of why such a pattern may be observed, nor of what has been observed of this in the past (e.g. Lasher-Trapp work)?

Page 13 line 18-19: "tendency for aerosols to suppress precipitation." Do you mean increase in aerosols?

Page 14 lines 12-14: Could it be that not all clouds reach the same altitude, and that the shallower ones thus bias the statistics at low altitude?

---

## Author Comment (AC1) · 10 Jun 2016

We greatly appreciate the thoughtful comments provided by the reviewers. We have responded to each comment of each reviewer (blue) outlined below and have modified the manuscript in response to these comments (please see the attached Supplement). The compressed file (as .zip) is attached that includes (1) Reply to reviewers' comments (referee #1 and #2) and (2) manuscript that shows the track changes.

[Figure]

Please also note the supplement to this comment:
http://www.atmos-chem-phys-discuss.net/acp-2016-45/acp-2016-45-AC1-supplement.zip

---

## Author Response (AR1)

See attched pdf file for my comments on this manuscript.
Please also note the supplement to this comment:
http://www.atmos-chem-phys-discuss.net/acp-2016-45/acp-2016-45-RC1-supplement.pdf

Review on
"*Aerosols, Clouds, and Precipitation in the North-Atlantic Trades Observed During the Barbados Aerosol Cloud Experiment. Part I: Distributions and Variability*" by Jung et at.

**General Comment:**
This paper presents a nice overview of „aerosol, cloud, and precipitation features" as measured over Barbados. Airborne data has been sampled in-situ combined with a cloud radar during 12 research flight. My overall opinion is that this paper has the clear motivation to characterize the observations without going into much detail of individual findings. I have no concerns about this strategy, in particular because the observation period nicely covers the three typical aerosol types typically observed at Barbados. However, at some points (see specific comments) a somewhat deeper analysis and discussion of individual findings would improve the paper instead of referring to another promised upcoming paper. However, I have a few major critical points which have to be seriously discussed before I can suggest this paper for publication.

1.) **Page 5, beginning of sec 2.5:** I have serious doubts about this method of simply combining two size distributions with one of it is sampled under ambeient and the other under dry conditions. The two size distributions may line up well but this could be really by chance! A careful analysis of this issue is absolute necessary. A lot of bigger aerosol might be sea salt particles which are highly hygroscopy and there should be a big difference if you measure it under dry or ambient conditions. Please include at least error bars which describe this effect! This is a serious point which needs more careful discussion.

In the revised manuscript, DSDs are not combined. Additionally, discrepancy occurring in ranges where two probe measurements overlay is discussed further in the text (in section 3.3.2).

2.) **Page 9, Sec 3.3.2:** Do I correctly understand? The PSD is averaged over all flight height for one individual flight - including sub-cloud layer and cloud layer heights? Does this make sense? Why not at least distinguish between sub-cloud layer and cloud layer aerosol?

I have serious doubts about the representativiness of these size distributions. In particular for the situation where dust is advected the size distribution should be a strong function of height.

Furthermore, all size distributions show more or less exactly the same general structure/shape with characteristic peaks and shoulder at the same size. It looks like that the distributions differ only due to

dilution although you mentioned that you had three different types of aerosol loading? Although the y-axis is logarithmic this seems quite strange. I discussed this issue with an expert for OPC measurements and we both wonder if this could be an instrument artefact. Please discuss this issue in detail.

Figure 6 shows PSD calculated from PCASP (integrated aerosols in the boundary layer), whereas Fig. 7 shows DSD combined from PCASP with CAS at sub-cloud layer.

PSDs in Fig. 6 were calculated from all available PCASP measurements made on pseudo-soundings and level flights for a given day, *when no liquid water was detected*, to give daily flight-averaged PSDs (L14-15, p.9). Therefore, PSDs shown in Fig. 6 does not include DSD from clouds.

We agree that the DSD should be a strong function of height especially when dust is advected (i.e., during the transition period). The manuscript clarifies the purpose of Fig. 6 is to show the daily flight-averaged PSDs. It shows that the overall shapes of PSD are similar during the BACEX **except for the transition period**. When we examined the individual size distributions there was plenty of variability in the DSD shapes. But "flight duration averages (i.e., flight-averaged PSD)" all seem to have the same shape in the boundary layer, only different concentration. Maybe there is some steady state distribution of particles on the average pretty general in the boundary layer, and the time-scale of the determining processes short enough that the shapes get to this equilibrium state fairly fast. During pre- and post-dust periods, a single type of aerosol predominates (maritime aerosol or dust). In contrast, dust particles begin to appear/disappear during the transition periods. This may be why two distinctive DSD shapes are evident during the transition periods (In the manuscript, DSDs of 29 March and 30 March were exampled).

PSDs in Fig. 6 indicate the integrated particle size distribution in the boundary layer. Therefore, the shapes of PSD in Fig. 6 reflect the overall concentrations/components of the aerosols in the boundary layer (single or multiple sources of aerosols). Figure 7, on the other hand, shows the DSDs obtained from sub-cloud layers. Hence, the PSDs in Fig. 7 reflect the effects of aerosol types at particular height (in this case, sub-cloud layer) on the shape of PSD. PSDs in Fig. 7 clearly show the different shapes of PSDs that depend on the predominant aerosol types.

In a revised manuscript, PSD in the sub-cloud layer is added in Fig. 6b to show the differences in PSD with height (integrated PSD in Fig. 6a versus PSDs in sub-cloud layer in Fig. 6b). Figure 6 shows the differences in PSDs for dusty and non-dusty days.

[Figure]

**Figure 6.** Daily averaged aerosol particle size distributions (PSDs) that obtained from the PCASP for (a) all cloud-free flights and (b) sub-cloud layer flights. Color bar indicates the research flight number (RF #), shown in Table A1. PSDs from the odd (even) RF numbers are shown as solid (dashed) lines. PSDs estimated between RF07 and RF10 (3/29, 3/30, 3/31, 4/1) are denoted as bold lines. PSDs of RF01 (3/19) and RF03 (3/23) are not shown due to the instrument malfunction (RF01) and the absence of PCASP data (RF03) for these days.

In the revised manuscript, DSDs are not combined from PCASP and CAS. Instead both DSDs are overlaid. DSDs shown in the bold lines are obtained from PCASP.

[Figure]

**Figure 7.** Daily averaged aerosol particle size distributions (PSDs) for the sub-cloud level flights. PSDs obtained from PCASP (0.1 $\mu$m to 2.5 $\mu$m) and CAS (0.6 $\mu$m to 60 $\mu$m) probes are shown as bold and thin lines, respectively. The color bar indicates the research flight number (RF #) presented in Table A1. PSDs from the odd (even) number of RF are shown as solid (dashed) lines. PSD of RF01 (19 March) and RF03 (23 March) are not shown due to the instrument malfunction and the absence of PCASP data for the days.

3.) **Page 16 (Summary):** This section is quite often a word-by-word summarize of the previous sections. The discussion is missed out completely. I totally understand that for a data overview paper a discussion is not a trivial task but repeating all the obsersations word by word is not a convincing solution.

Revised as "summary and conclusions" since the section summarized the findings of the study. In addition the authors added a more general statement to this section at the end.

**Specific Comments:**
Page 2, line 3ff: How are your findings biased by the sampling strategy - is it natural to assume that one tries to hit the bigger clouds which bias the frequency of observed cloud types?

We don't believe the sampling strategy is biased to the bigger clouds. However, it is possible that the pilot tries to avoid strong updrafts or downdrafts, although strong updrafts/downdrafts are not common in the shallow cumulus cloud fields. We did not particularly try to target the larger clouds. We first select good cloud fields and then sampled them flying near the cloud-base, mid-cloud and cloud-top.

P2, l 15ff: One might mention at least a few open questions. "Cloud have to better understood" is a quite generic statement.
Changes were made in the text as follows: Recent studies indicate that these clouds are the cause of the largest uncertainty in tropical cloud feedbacks in the climate system (e.g. Bony and Dufresne, 2005; IPCC, 2013) and therefore the characteristics and distributions of their variability must be better defined.

P2, l 17: Don't forget Malkus' landmark papers here; although these experiments were smaller three nice papers came out of it.
References are added in the introduction and changes are made accordingly as well.

P2, l 31: Why is Barbados a perfect place for such studies?- Be more specific here.
The reasons are given in L24, p2 - L1, p3.

P 4, l 6: "PVM-100 water content" is not a parameter - please use liquid water content instead.
Revised as suggested.

P4, l 8: Provide location of the company, use „Inc" instead „inc".
Revised as suggested.

P4, l 10ff: You should mention at this point which instrument measures under environmental conditions and which one is a closed system which samples under dried conditions - it's important here!

Revised as follows in the manuscript: Particles in PCASP, CPC and CCN are analyzed after they are been passed into the instruments. In the process, they warm up and dry out, which is really only an issue as far as the PCASP is concerned. The other instruments (i.e., CPC and CCN counter) grow the particles by condensing a fluid onto them before sizing them. CAS and CIP measure the particle sizes in the ambient conditions where their sizes are not altered.

P4, title of sec 2.3: Better "stratification"? Large-scale would also imply a horizontal component - right?
Revised as Atmospheric vertical structures

P4, l 26: Better characterized/analysed instead of „defined" ?
Revised as "analyzed"

P5, l 10ff: I have serious doubts about this point: The two size distributions may line up well but this could be really by chance! A careful analysis of this issue is absolute necessary. A lot of bigger aerosol might be sea salt particles which are highly hygroscopy and there should be a big difference if you measure it under dry or ambient conditions. Please include at least error bars which describe this effect! This is a serious point which needs more careful discussion (see general comment).

The revised Fig. 7 (shown above) shows that there is a discrepancy in DSDs that were obtained from PCASP and CAS in the ranges where the two probes overlap (0.8-2.5 $\mu$m), which may reflect the larger sea salt particles being swollen (overestimate the sizes from CAS). However, the DSDs from both probes line up well when considering the overall trends of the DSD with sizes.

Moreover, the main point we show in Fig. 7 remains unchanged, which is "Fig. 7 shows two distinct populations of PSDs in the sub-cloud layer. First, PSDs from dusty days (RF07-RF12) have a significantly higher $N_a$ between 0.5 $\mu$m and 10 $\mu$m, compared with PSDs that were obtained from the non-dusty days". Additionally, the authors added Appendix C and Fig. C1 to support "The effect of measuring the size of dust versus salt, which have different refractive indices, is relatively small in PCASP". Please refer to replies given for the general comments too.

**Appendix C: The effect of refractive index in PCASP and CAS**

The PCASP is pretty insensitive to refractive index (RI), but the forward scatter probe (CAS) is very sensitive to refractive index for D < 10 $\mu$m (Fig. C1). In the calibration plots (Fig. C1), the horizontal lines define the channel boundaries, the points show actual calibrations, and the continuous lines show response curves for various refractive indexes estimated from theory. The authors used the response boundaries for approximately middle of the RI envelopes. Shape may also cause sizing uncertainty. Here we only measure pulse heights, and use calibration using spherical glass beads and PSL's, as well as theoretical estimates of the probe response, to invert the pulse heights to 'diameter'. For the range of RI for atmospheric particles, the inversion may error by a factor of two in this size range, which is a well-known problem with forward scatter techniques. In Fig. C1, the modeled instrument response curves are labeled as RI with corresponding materials, and the rest are calibrations, labels by date of the calibration and material used. For example, "130212_dos" (cyan filled circles in PCASP plot (lower panel in Fig. C1) shows a calibration on 12 Feb. 2013 using Di Octal Sebacate oil drops to excite the instrument. The calibration points should line up with the theoretical DOS curve, calibrated for RI = 1.42. For effects of shape, not much is known. Only at high RH can one be reasonably certain that the soluble particles are wetted and spherical, and perhaps with RI close to that for water. The authors believe generating meaningful error bars accounting for all possible error sources is a big undertaking, and separating out the growth factor due to RH only doesn't add much to the results. Data are available in the literature on growth factors for various types of aerosol particles. Obviously they differ for different types, and we have no determination of the type. Considering all these effects Feingold et al. (2006) estimated that the accuracy in retrieved drop effective radius is within ~ 20 %.

[Figure]

**Figure C1**. Calibration plot. The horizontal lines define the channel boundaries, the individual points show the actual calibrations, and the continuous lines show response curves.

P 6, l 6: You could include the equation (in text) for the LCL determination but not really necessary
The equation is not included in the text.

P 6, l 33: Is there a difference/bias between the radiosondes and TO-profiles for determing the inversion? You mentioned earlier the method how to determine the main inversion and here you say that the inversion was poorly defined? Please specify!

The inversion is defined by the same method.
In the case that the vertical temperature gradient is large, the inversion is well-defined, which is common for stratocumulus clouds. However, the vertical temperature gradient is often weak for cumulus clouds. The sentence is revised as "with weak inversion heights in most flights".

P8 , l22: In the plot you abbreviate super-saturation with "ss", here with „s" - be consistent.
Changes were made to "ss" as suggested.

P9, l 15: Do I correctly understand? The PSD is averaged over all flight heights for one individual flight - including sub-cloud layer and cloud layer heights? Does this averaging make sense? Why not at least distinguish between sub-cloud layer and cloud layer aerosol?
PSDs from the cloud-layer were not included for this calculation.

I have serious doubts about the representativeness of these size distributions. In particular for the situation where dust is advected the size distribution should be a function of height. Furthermore, all size distributions show more or less exactly the same general structure with characteristic peaks and shoulder at the same size. It looks like that the distributions differ only due to dilution although you mentioned that you had three different types of aerosol loading? Although the y-axis is logarithmic this seems quite strange. I discussed this issue with an expert for OPC and we both wonder if this could be an instrument artefact. This point has to be discussed in detail (see general point).

Please see the above responses (in general point)

P9, l 18: I do not agree with the statement that PSDs avageraged over sub-cloud and cloud layer can provide any insight in processes. You average over regions where different processes are dominant. I suggest re-wording.

Please note that PSDs in Fig. 6 (flight-averaged PSD) do not include PSDs from clouds. The PSD here represents the integrated PSD in the clear-air boundary layer.

P 9, l 32: See my comment about combining an aerosol size dristribution measured with a closed system (dried aerosol) and an open system (ambient conditions).

In a revised manuscript, DSDs were not combined. They are simply shown together (revised Fig. 7). The statement (p9, L32: The plots in Fig. 7 show two distinct populations of PSDs in the sub-cloud layer) is still valid in a revised Fig. 7.

P 10, l 27: I don't understand exactly what you mean with "precipitating clouds exhibit more organized mesoscale features"? Can you please specify!

The characteristics of individual cloud, such as cloud features with strong cloud cores, anvils, and cloud envelops, are more evident in Fig. 8(a-b) compared with the clouds in Fig. 8(c-d; non-precipitating clouds), which are referred as the organized mesoscale features.

P 11, l 6: About LWC measureemnts with the PVM-100A; it is known that the transfer function of the PVM-100A decreases with increasing droplet size (see Wendisch, Garrett, and Strapp: 2002). The droplets in trade wind cumuli have comparable big droplets due to the low number concentration and this underestimation of the PVM might be an issue. Although you might not have the droplet sizes available you should at least mention this issue because it could partly result in the bigger discrepancy be between measured and adiabatic value around cloud top region. Of course you could take the effective radius derived from the PVM-100A and estimate the transfer function based on this parameter – but maybe mentioning this problem is also fine.

Revised as suggested. The following is added in the text: In addition to the cloud entrainment/detrainment and precipitation processes attributing to the discrepancy between measured and adiabatic value, the transfer function of PVM-100A being decrease with increasing droplet size (e.g., Wendisch et al., 2002) could also partly result in the discrepancy between measured and adiabatic value around cloud top region.

P 11, l 9: I don't completely understand this criterium, does it really completely rule out the possibility of counting shattered droplets? Please explain in more detail.

Counting shattered droplets is inevitable problem for all airborne probe measurements. We don't believe the threshold completely rules out the possibility of counting shattered droplets. However, the chances will be reduced by not counting the large drops (by considering data of CIP volume concentration < 0.1 $cm^{-3}$) that mainly cause the shattering issues. Criteria $w > 1$ m s$^{-1}$ and LWC > 0.01 gm$^{-3}$ are used to consider the cloud core.

Furthermore, a few line below you mention CARRIBA_dry: In the "polluted" case of CARRIBA_dry (April 24th) the aerosol number concentration in the SCL was about 265 particles per cm^3 resulting in mean droplet number concentration of 80 droplets per cm^3 (measured with a PDI). In your plots you have aerosol number concentrations of 100 to 300 per cm3 in the SCL (see Fig 2c) (assuming air density about 1 kg/m^3 in the SCL) but resulting in 100 to 200 droplets per cm^3 between z = 500 and 1000 m) in Fig 10b. This means that about 75% of the aerosol particle will be activated?

The sub-cloud layer aerosols and cloud-layer $N_d$ in a cloud core ($w > 1$ ms$^{-1}$) are shown in the Appendix figure 1 for a few days of BACEX to show how many aerosol particles are activated.
First of all, Fig. B1 shows that aerosol particles activate more efficiently in clean environments for a given updrafts ($w > 1$ ms$^{-1}$). For example, aerosols on 10-11 Aprils (Condensation nuclei (CN) < 300 cm$^{-3}$) activated about 70 %, whereas aerosols on 29 March and 5 April (CN > 300 cm$^{-3}$ and 400 cm$^{-3}$, respectively) activated about 50-60 % at cloud bases.

Furthermore, your profile of N_d is at least slightly increasing with height! It is not very typical that droplet number concentration is increasing with height although processes such as secondary activation might play a role for a few cases. Please discuss this point and offer an explanation for this behaviour.

The manuscript is revised as follows: The composite of $N_d$ obtained from 12 flights during BACEX (shown in Fig. 11) are shown in Fig. 10b. $N_d$ during BACEX varies from ~ 0 to 400 cm$^{-3}$ and tends to increase with height (Fig. 10b). The maximum $N_d$ occurred just above cloud base as commonly thought, as well as, high above the cloud base showing the tendency of increasing $N_d$ with heights. The increasing $N_d$ with heights was also observed in several research flights during the RICO. Further, the breadth of DSDs (in Fig. 10b) was predicted by the inhomogeneous mixing (Lehmann et al., 2009) allowing droplets to experience different degrees of subsaturation (e.g., inhomogeneous droplets evaporation considered by Bewley and Lasher-Trapp, 2011). The low $N_d$ at high altitude (~ 2300 m) may be associated with entrainment mixing and wet scavenging due to precipitation. The increasing or nearly constant $N_d$ with heights for the couple flights of BACEX are shown in Appendix B (Fig. B1) to show how many aerosol particles are activated during the BACEX.

On a first glance this looks like an inconsistency to me and should be discussed in much more detail. For example, you could show a profile of N_droplets for one flight which can be directly compared to the aerosol profile of the SCL? Then you can estimate how many aerosol particles are really activated. Your data suggest that more than 70% (or so) of the aeroosol as measured in the SCL will be activated

Changes are made in the manuscript in section 3.4, and Figure and text are added in the Appendix B.

P 11, l 9: Again, the parameter is "LWC" and not "PVM-100 LWC", by the way, in the plots the liquid water mixing ratio plotted and not the LWC.
Revised as suggested.

P 11, l 10: Wasn't there a Phase Doppler Interferometer (PICT) installed? If yes and if it worked properly, you should have a reliable estimate of the droplet number concentration (and size distribution).
During BACEX, a PICT was not operated. Cloud droplet number concentrations are measured by Cloud Aerosol Spectrometer (CAS) in this study.

P 11, l 15: I don't understand the "comparison" with the CARRIBA_dry campaign in this context. The three differernt types of aerosol origin were also observed during CARRIBA.
Comparison with CARRIBAR_dry is made simply to address the fact that the BACEX was seasonally similar to the CARRIBA$_{DRY}$ period but observed the strongest dust event during all of 2010.

End of p 11: I am not an expert in radar meteorology but a liitle bit more detailed discussion would be appreciated. What can I learn from the radar measurements?
Changes were made in the text as follows: The radar measures returned signals (Z) of the object, which are proportional to D$^6$ and the number concentration of droplets (see eq. 3) and Doppler velocity (*Vr*) of the target which is moving toward or away from the radar. In general, the larger the hydrometer, the stronger Z is measured. Precipitating clouds are often Z > -17 to -20 dBz in cloud radars (e.g., Frisch et al., 1995).

P 12, l 6: From reading this part of the text it is difficult to understand the differences of the four patterns in Fig 12. - in particular between panel b) and c). How is "strongest precipitationd" defined and why do you exclude a few precipitating clouds in panel d) ? – On a first glance this seems to be a little bit arbitrary and the motivation for doing this remains unclear. A few more word would help the reader.

Changes were made in the text and Fig. 12 is revised.

FYI
Figure 12 shows that how the *V$_r$* and *Z* distributions differ from precipitating and non-precipitating clouds. In Fig. 12, there are two distinctive patterns that correspond to precipitating clouds (Fig. 12b) and non-precipitating clouds (e.g., Fig. 12d).

[Figure]

**An area of [A] in (c) corresponds to the precipitating cloud pattern, whereas [B] indicates the non-precipitating pattern

Precipitating clouds are shown as echoes that have maximum of strong radar returns (Z) and negative $V_r$ (hydrometer fall toward ground). For example, the most frequently observed echoes in Fig. 12b are Z ~-10 to -20dBz (which corresponds to the radar return of precipitating particles) and $V_r < 0$ (which indicates particles are falling down toward the ground). In Fig. 12d, the most frequently observed echoes are those of Z < -30 dBz (which corresponds to the echoes of non-precipitating particles) and $V_r$ ±2 m/s (particles have up and down motion, which corresponds to more likely random air motion). The manuscript is revised as followings "The $V_r$-Z distribution that is estimated by excluding the strongest precipitating cloud on 22 March (Fig. 12c) shows two populations of Z and $V_r$: first, weak Z (Z < -30 dBz) with wide ranges of $V_r$ (-4 m s$^{-1}$ < $V_r$ < 3 m s$^{-1}$), which is shown as an horizontally oriented pattern in Fig. 12c; and second, strong Z (e.g., -30 dBz < Z < 5 dBz) with predominately a negative $V_r$ ( 0 m s$^{-1}$ < $V_r$ < ~ -4 m s$^{-1}$), which is shown as an vertically oriented pattern in Fig. 12c".

P 12, l 22: Have you really excluded three precipitating clouds or three days with precip?
Three days with precipitating clouds. Revised the manuscript to clarify the point.

P12, l 24: I am confused here: a few lines above you claimed that clouds deeper 500 to 600 m have a significant chance of precipitating and now you analyze the non-precipitating clouds in panel c) and d) with a thickness of about 1300 m? isn't this contradictionary or do I misunderstood something? Please specify!

We agree that the definition of non-precipitating clouds here may be confusing.
In the manuscript (between page 11 and page 12) it is defined as Clouds sampled on these three days (3/22, 3/24, and 3/30) are referred to as "precipitating-clouds" hereafter, whereas clouds sampled on 9 days that excluded these three precipitating-cloud cases are referred to as "non-precipitating clouds" . Therefore, the non-precipitating clouds shown in Fig. 13 indicate clouds sampled during 9 days when no heavy precipitation (e.g., 3/22, 3/24, 3/30) is observed. However, these 9 clouds include clouds that do

not precipitate at all (purely non-precipitating, such as 3/25, 3/26, 3/29, 3/31, 4/11) as well as lightly precipitating clouds mainly from the cloud tops (e.g., 3/23, 4/5, 4/7, 4/10). These features are readily shown in Fig. 11 or Fig. 14.

Manuscript is revised to clarify it as follows: For these clouds (defined as non-precipitating clouds in this study, but in reality, the clouds could have light precipitation), $Z \sim$ -35 dBz and $V_r$ of $\pm$ 2 m s$^{-1}$ are the most frequently observed between 600 m and 1300 m. Cloud bases and tops for the clouds are about 700 m and 2000 m, respectively, indicating a thickness of about 1300 m.

P 13, l 10: One should also mention in the text that the photo in Fig 15 is not from BACEX
Added "Note that the photo is of a cloud over Key Biscayne, FL in an environment similar to that in Barbados" to the end of the paragraph.

P 13, l 15: The following part is more discussion and not data analysis. However, I do not see how this discussion is justified by the observed data and its analysis. This part is a little bit confusing to me and should be re-worded.
We removed this part since a similar discussion is given in the summary and discussion section.

P 13, l 27: Although there is a strong collaboration between the CARRIBA community and the activity ot the MPI group, the work of L Nuijens et al. should not be considered as part of CARRIBA.
Changes were made in the manuscript.

P 14, l 1: I think the frequency of precip should be mentioned earlier - it is a quite fundamental point and comes surprisingly late in this section.
The main purpose of Fig. 16 is to confirm the predominance of the second type of precipitation in the Barbados, and thus it may be proper to show together with Fig. 14-15. Therefore the order is not changed.

P 14, l 4: How is the precipitation LWC defined here?
It is defined as CIP volume concentration multiplied by the density of water.

About „Summary and Discussion" Overall comment: This section is really mostly a word-by word summarize of the previous observations. The discussion part is missed out completely. I totally understand that for a data overview paper a discussion is not a trivial task but repeating all the observations word by word is not a convincing solution and I suggest a complete re-wording of this section.

Revised as "summary and conclusions" since the section summarized the findings of the study. In addition the authors added a more general statement at the end.

P 14, l 32: It is a little bit strange to talk about hurricane seasons for Barbados because Barbados wasn't influenced by a hurricane for several decades (before your flights, the first hurricane „Tomas" came in just a few month later...
Hurricane season is a commonly used terminology for the hurricane community that indicates the period of July to September. We changed hurricane seasons to July to October in the manuscript.

P 15, l 2ff: This sentence is quite confusing because one can interpret it such as the maximum of N_a is below the inversion which of course makes no sense if it is steadily decreasing with height - I suggest re-wording to make it clear what you mean.

Changes were made in the text by removing "below the trade wind inversion".

P 15, l 12: Just a comment but there are very contradicionary studies about a possible influence of GCCN on the development of precipitation - could be at least mentioned here.

Changes were made in the text to clarify it.

P 15, l 16: What do you mean with "could provide" I think at this point a deeper data analysis would be interesting.

This topic is addressed in Jung et al., 2013 using the African dust events (for example, Jung et al. (2013) showed that cloud (history of cloud) processes in boundary layer caused complicated stratification in the aerosols below the Saharan Air layer).

P 15, l 24: I think a little bit more quantitative analysis would be justified at this point! That cumulus clouds are far away from being adiabatic is well known.

It is well-known that the cumulus clouds are far from being adiabatic. The points that we want to make here are that the adiabatic assumption that is commonly used in satellite studies needs to be considered carefully. Changes were made to clarify this point.

Figures:

Fig 1: to include the units in brackets is confusing because one would expect the parameter which was mentioned before - the potential temperature is "\Theta" and not "K" – just a formal point.

Units are removed from the figure.

Fig 1: Although I would do it in the same way: is there a good argument for taking the readings in the height range between 100 and 200 m? I assume the radiosonde takes some time to provide reliable readings? If you would take data from the TO you could take the 30m level legs - right?

We kept the figure because we believe there is no good argument for taking the reading in the height between 100 and 200 m for the purpose of Fig. 1 is to show the overall structure of atmospheric conditions (in the entire boundary layer).

Fig 2: Although it is a nice overview, details of the profiles are difficult to detect - this is particular true for \Theta and q_l.

We agree that the details of the profiles are difficult to detect. However, the purpose of the plot is to show the overall thermodynamic structures in the boundary layer during the experiment and the variability. Nevertheless, the profiles show two distinctive thermodynamic structures for the dusty and non-dusty days. It is shown in P7 L3-4 "The overall atmospheric conditions and variability observed from the TO are illustrated in Fig. 2 with the vertical profiles of potential temperature ($\theta$), mixing ratio ($q$), and aerosol concentration ($N_a$)"

Fig 2c: The unit „particles per mg" - a kind of aerosol number mixing ratio - is not very common (although it makes sense...). Usually I would expect particles per cm^3 normalized to standard conditions - please specify in the text what you did here.

The unit of [$\#/cm^3$] does not count for the effect of density vriability. [#/mg] represents #/cm3 divided by the density to account for the density effect with heights.

Fig 2a: In my pdf copy the "\Theta" does not show up.

[Figure]

**Fig. 2.** Profiles of (a) potential temperature, Ѳ, (b) water vapor mixing ratio (g/kg), and (c) aerosol number concentration per mass of air (#/mg) obtained from PCASP during the aircraft's ascents and/or descents. The profiles shown are one out of many soundings for each day and are denoted in Table A1. The color bar shows the number of research flight (RF #), shown in Table A1.

Fig 2 caption: Why do you use sometimes symbols, sometimes not? I suggest to use symbols in the xlabels of the plot and in the caption I suggest using "water vapor mixing ratio q_l(g/kg)" and being consistent with this notation throughout the paper.

$r_v$ and $r_l$ is used for the mixing ratio and liquid water mixing ratio in the revised manuscript (Fig. 2, Fig. 3, Fig. 10).

Fig 3: You could consider plotting the 1-sigma as classical error bars or at least dotted lines. First I was confused because it looks like another data set but you have described it in the figure caption so everything is technically correct.
Everything is technically correct.

Fig 5: Difficult to figure out the absolute values.

[Figure]

Fig. 5(a): The average (black-solid lines) and individual (colored lines) profiles of $N_a$ (m g$^{-1}$) are offset by 400 mg$^{-1}$ for each flight in Fig. 5a, *with each vertical dotted line representing a new axis to indicate aerosol concentration for the day in question*. For example, $N_a$ on 5 April is nearly constant below 500-600 m (~ 300 mg$^{-1}$), then gradually increases with height and peaks around 2000 m at ~ 600-700 mg$^{-1}$. Thereafter $N_a$ decreases with height reaching ~ 200 mg$^{-1}$ at around 2500 m.

Fig 5, ylabel: Is here a "micron" missing? it does not show up in my pdf copy? Please check all figures for this problem.
The units are height (m) in Fig. 5a, (#/mg) in Fig. 5b and ($\mu$g/m$^3$) in Fig. 5c.

Fig 9: The red dashed lines should be a little bit thicker!
Revised as suggested.

Fig 10: From this profiles it looks like that cloud base is 250 m below LCL ? and around cloud base you have super-adiabatic q_l? I don't understand the calculation of adiabatic q_l; is it estimated for each cloud core individually? If so, why do you have super adiabatic q_l between mean LCL and minimum LCL? Why not normalizing each cloud with its adiabatic value or showing the difference between actual q_l and adiabatic value for each cloud?

It is a composite of 12 clouds (that are shown in Fig. 11), where the clouds are sampled in updraft regions (w > 1 ms-1). Please note that the cloud is not normalized by the cloud thickness. However, the cloud base heights for each day are similar except for one flight (24 March), and the samples that look like super-adiabatic were obtained from 24 March. The figure is revised.

[Figure]

**Fig. 10.** (a) Cloud liquid water mixing ratio ($r_l$) and (b) droplet number concentration $N_d$ in cloud core ($w > 1$ m s$^{-1}$) sampled by the Twin Otter during BACEX (12 flights shown in Fig. 11). Data points (greys in Fig. 10a and colors in Fig. 10b) are averaged in 10 m (vertical) for all clouds (red in Fig. 10a, black in Fig. 10b) and for precipitating clouds (blue in Fig. 10). $N_d$ from the individual flights are shown as colors that shown in Fig. 2. The mean (minimum and maximum) values of LCL are denoted by dashed (dotted) lines.
Review of: Aerosols, Clouds, and Precipitation in the North-Atlantic Trades Observed During the Barbados Aerosol Cloud Experiment. Part I: Distributions and Variability By Jung, Albrecht, Feingold, Jonsson, Chuang and Donaher.
Evaluation: Publish with minor revisions

Major comments:
This very much reads as a paper setting the stage for something to come. As such it is a bit light, but it does contain sufficient information to warrant publication.

Page 5 line 11: "The PCASP dries the particles before measuring them." Please provide a reference for this. For instance, Strapp et al (1992, JAOTech) leaves the door open for the PCASP only partially drying particles larger than sub-micrometer (their summary). Thus the behavior may be quite different for a dust particle (or for a coated dust particle) and a sea-salt particle. Other references?

The reference is added and the manuscript is revised in the beginning of the second paragraph in section 3.3.2.

Page 5 line 15: Missing integration sign in denominator.
Integration sign is added.

Page 9 lines 24-26: I am not sure what you are implying here; does sea-salt not contribute to the larger sizes in Fig. 6? This would seem to be inconsistent with many studies that have found sea-salt in the entire range of marine aerosols, e.g. Clarke et al. (2006, JGR), Blot et al (2013, JGR), Modini et al (2015, JGR).
The last part of the sentence was removed to clarify it. What we intended here was GCCN contributed to the larger sizes, yet the concentration of GCCN in nature is small ($10^{-2}$-$10^{-4}$) compared with dust concentrations.

Page 10, lines 8-9: A reference for the sizing uncertainty?
Appendix C and Figure C1 are added in the manuscript.

Page 31 and rest of manuscript: Are you connecting a PCASP (which you claims dries particles), with a CIP which does not materially dry dust particles (except maybe for a thin coating layer) and which when measuring sea-salt will see un-dried hydrated particles. I do not see any discussion of the fact that you do not necessarily know what is being looked at with the CIP; maybe I missed it?

In a revised manuscript, two probes (either PCASP and CIP or PCASP and CAS) were not combined (Fig. 7). Further, discussion is added for the issue in section 3.3.2. Please refer to the supplementary material.

Page 11 line 9: What is a "CIP volume number concentration"? It occurs several times.
It should be CIP volume concentration, which is proportional to the mass. Revised throughout the manuscript.

Page 11 line 20: "increases with height." This is one of the places that the manuscript comes up missing some context. No discussion of why such a pattern may be observed, nor of what has been observed of this in the past (e.g. Lasher-Trapp work)?

The manuscript is revised as follows: The composite of $N_d$ obtained from 12 flights during BACEX (shown in Fig. 11) are shown in Fig. 10b. $N_d$ during BACEX varies from ~ 0 to 400 cm$^{-3}$ and tends to increase with height (Fig. 10b). The maximum $N_d$ occured just above cloud base as commonly thought, as well as, high above the cloud base showing the tendency of increasing $N_d$ with heights. The increasing $N_d$ with height was also observed in several research flights during the RICO. Further, the breadth of DSDs (in Fig. 10b) was predicted by the inhomogeneous mixing (Lehmann et al., 2009) allowing droplets to experience different degrees of sub-saturation (e.g., inhomogeneous droplets evaporation is considered by Bewley and Lasher-Trapp, 2011). The low $N_d$ at high altitude (~ 2300 m) may be associated with entrainment mixing and wet scavenging due to precipitation. The $N_d$ with heights for the couple flights of BACEX are shown in Appendix B (Fig. B1) to show how many aerosol particles are activated during the BACEX.

Page 13 line 18-19: "tendency for aerosols to suppress precipitation." Do you mean increase in aerosols?

In general, precipitation is suppressed as aerosol concentrations increase in warm marine boundary layer. The detrainment moistening and evaporative cooling near cloud top can destabilize the local environment and promote deeper clouds (regardless of the aerosol loading), and the deeper and wetter clouds would tend to precipitate more (even if the initial environments has high concentration of aerosols), which can offset the tendency for aerosols to suppress precipitation.

Page 14 lines 12-14: Could it be that not all clouds reach the same altitude, and that the shallower ones thus bias the statistics at low altitude?

The statistics may result from sampling more (in number) shallower clouds than deeper clouds. However, it could be the nature of the clouds such that more abundant shallower clouds exist at a given moment than deeper clouds.

[revised manuscript text omitted]